# Directional Textual Inversion for Personalized Text-to-Image Generation

**Kunhee Kim**[1*]**, NaHyeon Park**[1*]**, Kibeom Hong**[2] **& Hyunjung Shim**[1]
[1]KAIST, [2]Sookmyung Woman's University
{kunhee.kim,julia19,kateshim}@kaist.ac.kr
kb.hong@sookmyung.ac.kr

## Abstract

Textual Inversion (TI) is an efficient approach to text-to-image personalization but often fails on complex prompts. We trace these failures to embedding norm inflation: learned tokens drift to out-of-distribution magnitudes, degrading prompt conditioning in pre-norm Transformers. Empirically, we show semantics are primarily encoded by direction in CLIP token space, while inflated norms harm contextualization; theoretically, we analyze how large magnitudes attenuate positional information and hinder residual updates in pre-norm blocks. We propose Directional Textual Inversion (DTI), which fixes the embedding magnitude to an in-distribution scale and optimizes only direction on the unit hypersphere via Riemannian SGD. We cast direction learning as MAP with a von Mises-Fisher prior, yielding a constant-direction prior gradient that is simple and efficient to incorporate. Across personalization tasks, DTI improves text fidelity over TI and TI-variants while maintaining subject similarity. Crucially, DTI's hyperspherical parameterization enables smooth, semantically coherent interpolation between learned concepts (slerp), a capability that is absent in standard TI. Our findings suggest that direction-only optimization is a robust and scalable path for prompt-faithful personalization. Code is available at https://github.com/kunheek/dti.

## 1 Introduction

Personalization in text-to-image generation involves the targeted adaptation of models to learn representations of novel, user-provided concepts (Gal et al., 2023a; Ruiz et al., 2023). This process allows for the creation of customized images that faithfully render specific concepts, such as unique individuals, objects, or artistic styles, in new contexts.

Current personalization approaches fall into two paradigms: parameter fine-tuning and embedding optimization. Parameter fine-tuning methods, exemplified by DreamBooth (Ruiz et al., 2023), optimize entire models using a few user-provided images. While effective, these approaches are computationally expensive and require significant storage per concept. In contrast, embedding optimization methods, such as Textual Inversion (Gal et al., 2023a), offer a more efficient alternative by optimizing only token embeddings. This approach provides substantial advantages: minimal storage per concept and seamless workflow integration. These advantages have made TI a foundational component in numerous personalization frameworks (Hao et al., 2023; Kumari et al., 2023; Tewel et al., 2023; Lee et al., 2024) and align with a broader paradigm shared with other domains, such as LLMs (Lester et al., 2021) and VLMs (Alaluf et al., 2024).

Despite its utility, TI suffers from critical limitations. The fundamental challenge stems from the restrictive constraint of optimizing a single embedding vector to encapsulate highly complex and multifaceted visual concepts. This limitation leads to two key problems. First, TI struggles to maintain high fidelity to complex prompts, compromising its controllability and expressive range. Second, the extensive fine-tuning duration required to learn each new concept hinders its practical applicability for rapid, user-driven workflows. Recent works (Voynov et al., 2023; Alaluf et al., 2023) have attempted to address these limitations through enriched embedding spaces, but introduce significant

---

*Equal contributions.

computational overhead that undermines TI's efficiency advantage. Moreover, these existing methods merely treat the symptoms rather than directly addressing the underlying optimization dynamics of TI, leaving the fundamental geometric factors that govern semantic alignment in embedding-based personalization largely unclear.

To bridge this gap, this paper presents a systematic, interpretability-driven analysis of the optimization dynamics inherent to TI, with a specific focus on the geometric characteristics of the token embedding space. By carefully dissecting the learned representations, our investigation reveals that semantic information is predominantly encoded in the direction of the embedding vectors. Furthermore, we demonstrate both theoretically and empirically that the unconstrained magnitude of these embeddings is a primary source of instability; specifically, excessively high embedding norms emerge during optimization and act as a critical factor impairing image-text alignment.

Building on these findings, we introduce **Directional Textual Inversion (DTI)**, a novel framework designed to address these fundamental limitations. Unlike conventional methods that optimize the entire token embedding, DTI decouples embeddings into their magnitude and directional components. Our approach maintains the embedding magnitude at a scale consistent with in-distribution tokens from the pre-trained model, while focusing the optimization exclusively on the embedding's direction. To enhance semantic coherence, we formulate this directional optimization as a Maximum a Posteriori (MAP) estimation problem. This formulation incorporates a von Mises-Fisher (vMF) distribution as a directional prior, which effectively regularizes the embedding towards semantically meaningful directions in the hyperspherical latent space. The resulting framework preserves the lightweight nature of TI while significantly improving its robustness, ensuring that personalization is both computationally efficient and semantically faithful.

Our comprehensive evaluation demonstrates that DTI consistently outperforms conventional TI and existing enhancement methods such as CrossInit (Pang et al., 2024a), achieving substantial improvements in semantic fidelity while maintaining computational efficiency. Beyond performance gains, the directionally optimized embeddings also enable novel applications, especially smooth interpolation between personalized concepts, expanding creative possibilities in generative AI workflows.

## 2 ANALYZING TOKEN EMBEDDING GEOMETRY

This section examines the token embedding space of pre-norm Transformer architectures, such as the CLIP text encoder (Radford et al., 2021) and Gemma (Team et al., 2024), which are foundational to modern text-to-image models. Our analysis establishes two key findings. First, we demonstrate that semantic information is primarily encoded in the direction of an embedding vector. Second, we identify that an excessively large embedding magnitude is a common artifact of standard Textual Inversion, a phenomenon we show is detrimental to model performance. We substantiate these findings with empirical observations and subsequently develop a theoretical framework to elucidate the underlying cause.

### 2.1 EMPIRICAL MOTIVATION: DIRECTION ENCODES SEMANTICS

Our first observation is that the semantic structure of the textual token embedding space is predominantly directional. This aligns with the foundational principle of semantic vector spaces where meaning is encoded not in the vector's magnitude, but in its direction (Mikolov et al., 2013; Pennington et al., 2014). We empirically demonstrate this by comparing nearest neighbors for a given token using two different distance metrics: Euclidean distance, which is sensitive to both magnitude and direction, and cosine similarity, which is sensitive only to direction.

Table 1: Top 5 nearest tokens to 'apple' under different measures.

| Rank | Euclidean | Cosine |
|------|-----------|--------|
| 1 | U+2069 | apples |
| 2 | altrin | fruit |
| 3 | lestwe | peach |
| 4 | heartnews | pear |
| 5 | samanthaprabhu | egg |

The superior semantic coherence of neighbors found using cosine similarity validates the principle that meaning in these vector spaces is encoded primarily by direction.

As shown in Table 1, an embedding's nearest neighbors are semantically coherent when measured by cosine similarity but not by Euclidean distance. For the token 'apple', its cosine-based neighbors include 'apples', 'fruit', and 'pear', while its Euclidean-based neighbors are often semantically

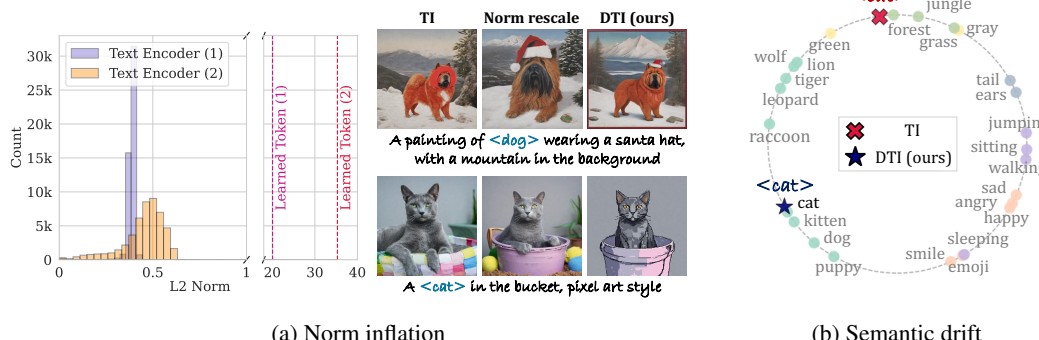

(a) Norm inflation            (b) Semantic drift

Figure 1: Empirical motivation for our method. Our analysis reveals two critical problems in standard TI that degrade prompt fidelity. (a) TI produces embeddings with excessive norms compared to model's original vocabulary. (b) TI also suffers from semantic drift, where learned embedding direction moves away from related concepts. These observations motivate DTI, an approach designed to preserve both norm and directional integrity.

unrelated tokens that merely share a similar magnitude. This indicates that an embedding's direction is the primary carrier of semantic information. More results are provided in Appendix A.

Figure 1b further illustrates this principle, showing that related concepts are located proximally on the unit hypersphere. Despite this, standard TI often neglects the importance of direction. This oversight leads to semantic drift, where the learned embedding for a token like <cat> moves directionally away from related concepts like 'cat' and 'kitten', as shown in the figure. This deficiency motivates the need for a method that explicitly preserves the semantic direction of learned embeddings.

## 2.2 WHY LARGE MAGNITUDES LEAD TO LOW TEXT FIDELITY

As shown in Figure 1a, TI produces token embeddings with norms that are drastically larger than those of the pre-trained vocabulary (often $> 20$ vs. $\approx 0.4$). These out-of-distribution (OOD) magnitudes consistently correlate with poor prompt fidelity. For instance, a prompt like "A painting of <dog> wearing a santa hat" may generate the dog but omit the hat and background details. While simply rescaling the embedding's norm after training can partially recover text alignment, it does not solve the underlying issue and can degenerate subject similarity. This raises a critical question: why do large embedding norms degrade text fidelity in pre-norm Transformers?

Our analysis reveals two primary mechanisms through which large-norm embeddings disrupt the Transformer's ability to contextualize information. We analyze a standard pre-norm Transformer block, $\boldsymbol{y} = \boldsymbol{x} + F_\ell(\mathrm{Norm}(\boldsymbol{x}))$, where $\mathrm{Norm} \in \{\mathrm{LayerNorm}, \mathrm{RMSNorm}\}$ and $F_\ell$ denotes attention/MLP sub-layers. We decompose the learned token as $\boldsymbol{x}^{(0)} = m\,\boldsymbol{v} + \boldsymbol{p}$ with $m > 0$ (magnitude), $\|\boldsymbol{v}\|_2 = 1$ (direction), and an additive positional embedding $\boldsymbol{p}$. Below, we explain how a large magnitude $m$ undermines the model's performance. (For formal proofs, see Appendix B).

**Effect I: Positional information is attenuated (see Lemma 1).** After LayerNorm/RMSNorm layer, the normalized signal that feeds attention/MLP becomes less sensitive to small additive terms as $m$ grows. Positional information contributes $\mathcal{O}(1/m)$ to the normalized signal $\mathrm{Norm}(m\boldsymbol{v} + \boldsymbol{p})$. Intuitively, a very large-norm token *forgets where it is in the sequence*, weakening contextualization, resulting in omission of details such as style and background (see Figure 1).

**Effect II: Residual updates stagnate (see Lemma 2).** The residual updates, $F_\ell(\mathrm{Norm}(\boldsymbol{x}^{(\ell)}))$, are computed from *normalized* inputs and thus have a bounded magnitude. When this bounded update is added through the skip connection to a large vector $\boldsymbol{x}^{(l)}$, the *relative* change (i.e., turning angle of the hidden state's direction) becomes tiny, decreasing in proportion to $1/\|\boldsymbol{x}^{(l)}\|$. In other words, large-norm hidden states become *stuck* in their direction and are difficult for subsequent layers to refine. This *residual stagnation* accumulates across layers, severely limiting the total directional change the initial token can undergo, as formalized in the following proposition and corollary.

**Proposition 1** (Accumulated directional drift across $L$ pre-norm blocks). *Let $\boldsymbol{x}^{(0)} \neq \boldsymbol{0}$ and $\boldsymbol{x}^{(\ell+1)} = \boldsymbol{x}^{(\ell)} + F_\ell(\mathrm{Norm}(\boldsymbol{x}^{(\ell)}))$ for $\ell = 0, \ldots, L-1$. Let $B_\ell := \sup_{\boldsymbol{u} \in S} \|F_\ell(\boldsymbol{u})\|_2 < \infty$, and $S_L := \sum_{j=0}^{L-1} B_j$. Assume $\|\boldsymbol{x}^{(0)}\|_2 > S_L$, then*

$$\angle\big(\boldsymbol{x}^{(0)}, \boldsymbol{x}^{(L)}\big) \ \leq \ \frac{\pi}{2} \sum_{\ell=0}^{L-1} \frac{B_\ell}{\|\boldsymbol{x}^{(0)}\|_2 - \sum_{j<\ell} B_j} \ \leq \ \frac{\pi}{2} \frac{S_L}{\|\boldsymbol{x}^{(0)}\|_2 - S_L}.$$

**Corollary 1** (Scaling $\Rightarrow$ directional freezing). *With the notation of Proposition 1, for any $\alpha > 1$,*

$$\angle(\alpha\boldsymbol{x}^{(0)}, \boldsymbol{x}^{(L)}(\alpha)) \ \leq \ \frac{\pi}{2} \frac{S_L}{\alpha \left\|\boldsymbol{x}^{(0)}\right\| - S_L} \ \xrightarrow{\alpha \to \infty} \ 0,$$

*where $\boldsymbol{x}^{(L)}(\alpha)$ denotes the depth-$L$ output when the initial token is $\alpha\boldsymbol{x}^{(0)}$.*

Together, these two effects explain why TI struggles with text fidelity. As a token's magnitude increases, its ability to integrate contextual information from the prompt diminishes. The personalized token becomes so dominant that it overshadows other critical details, such as stylistic elements, background context, or additional subjects, from the generated output. To this end, this analysis highlights the need for a method that explicitly controls the magnitude of personalized tokens, which we introduce in the next section.

## 2.3 EMPIRICAL VALIDATION

We empirically validate the two theoretical effects introduced in the previous sections. Effect I describes the attenuation of positional information under large embedding magnitudes, while Effect II concerns residual-update stagnation in pre-norm Transformer blocks. Our experiments directly probe both behaviors on the base encoder, TI, and our proposed DTI.

**Effect I (Attenuation of positional information).** We evaluate whether increasing embedding magnitude makes positional information unrecoverable after the first pre-norm normalization (LN). To validate this, we train a 2-layer MLP classifier on the *frozen* base text encoder to predict a token's absolute position from the output of $\mathrm{LN}(\mathbf{e} + \mathbf{p})$, where $\mathbf{e}$ and $\mathbf{p}$ each denote token and positional embeddings. On unmodified inputs ('Normal' in Figure 2), the classifier achieves $100\%$ accuracy, confirming that LN preserves positional information. We then scale the norm of a single token embedding by a factor $m \in \{0.5, 1, 2, 4, 8, 16\}$

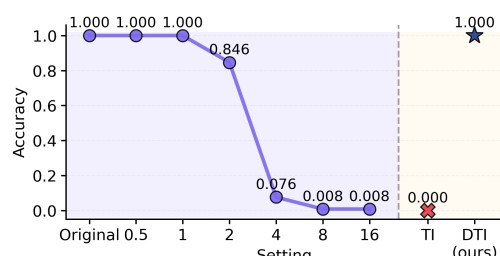

Figure 2: Position prediction accuracy from LN outputs under varying embedding magnitudes, compared with trained TI and DTI embeddings.

before applying LN. Accuracy deteriorates rapidly once $m$ exceeds the natural scale of the encoder. Furthermore, we evaluated the classifier on TI-trained and DTI-trained personalized embeddings. TI embeddings, which have excessively large norms, collapse to near-zero positional accuracy, while DTI embeddings remain fully recoverable.

This behavior directly corroborates Lemma 1: when $m$ becomes large, $\mathrm{LN}(m\mathbf{v} + \mathbf{p})$ becomes dominated by $m\mathbf{v}$, rendering the positional component $\mathbf{p}$ effectively invisible. DTI avoids this failure mode by constraining magnitudes to remain in-distribution.

**Effect II (Residual-update stagnation).** To test Lemma 2, we measure the internal angular change of hidden states within each pre-norm Transformer block. For each concept token, we compute the angle between the hidden state entering and exiting each block and then average across all layers. The average per-block angular change of TI embeddings was $21.33°$, whereas the angular change of DTI embeddings was $33.52°$ ($\mathbf{1.57\times}$ larger).

These results support the theoretical prediction that excessively large norms suppress the effective residual direction in pre-norm blocks, causing the forward computation to behave nearly as an identity mapping. By keeping embedding norms within the training distribution, DTI prevents such stagnation and permits substantially larger and more meaningful updates throughout the encoder.

## 3    METHOD: DIRECTIONAL TEXTUAL INVERSION

Based on the observation and analysis in the previous section that token embeddings exhibit strong directional characteristics, we introduce *Directional Textual Inversion* (DTI), a framework that optimizes an embedding's direction with in-distribution norm to enhance text fidelity in personalized text-to-image generation.

### 3.1    OPTIMIZING ONLY DIRECTION ON THE HYPERSPHERE

We reformulate TI by decoupling the magnitude and direction of the learnable token embedding $e \in \mathbb{R}^d$. The embedding can be expressed as

$$e = m^\star v, \qquad v \in \mathbb{S}^{d-1}. \qquad (1)$$

Here, $\mathbb{S}^{d-1} = \{u \in R^d : \|u\|_2 = 1\}$ denotes the unit sphere. We fix the magnitude $m^\star$ and optimize only the direction ($v$). Specifically, we set $m^\star$ to be an *in-distribution* magnitude derived from the frozen vocabulary of the text encoder (e.g., the average norm). In this way, optimization focuses on semantic information in direction while avoiding out-of-distribution (OOD) norms.

---

**Algorithm 1** Directional Textual Inversion (DTI)

1: **Inputs:** Model $\epsilon_\theta$, text encoder $c(\cdot)$, init token $e_{\text{init}}$, magnitude $m^*$, $\kappa$, iterations $K$, learning rate $\eta$
2: $v_0 \leftarrow e_{\text{init}}/\|e_{\text{init}}\|_2$
3: $\mu \leftarrow e_{\text{init}}/\|e_{\text{init}}\|_2$
4: **for** $k = 0$ to $K - 1$ **do**
5:     Sample minibatch $(z, t, \epsilon)$
6:     $g_{\text{data}} \leftarrow \nabla_v \mathcal{L}_{\text{data}}(m^* v_k)$
7:     $g_{\text{euc}} \leftarrow g_{\text{data}} - \kappa \mu$     (add prior gradient)
8:     $g \leftarrow g_{\text{euc}} - (g_{\text{euc}}^{\mathsf{T}} v_k) v_k$   (tangent projection)
9:     $g' \leftarrow g/\|g\|_2$        (gradient scaling)
10:    $v_{k+1} \leftarrow \dfrac{v_k - \eta g'}{\|v_k - \eta g'\|_2}$   (retraction to $\mathcal{S}^{d-1}$)
11: **end for**
12: **return** $e^* = m^* v_K$

---

Since the parameter space is the unit sphere, Euclidean updates drift off-manifold, rendering AdamW (Loshchilov & Hutter, 2019)–the default optimizer for TI-like methods–unsuitable. To solve this, we use Riemannian stochastic gradient descent (RSGD) (Bonnabel, 2013) with tangent-space projection and retraction:

$$g = g_{\text{euc}} - (v_k^{\mathsf{T}} g_{\text{euc}}) v_k \in T_{v_k}\mathbb{S}^{d-1}, \quad v_{k+1} = \text{Retr}_{v_k}(-\eta g) = \frac{v_k - \eta g}{\|v_k - \eta g\|_2}. \qquad (2)$$

Here, $g_{\text{euc}}$ is a Euclidean space gradient, $g \in T_{v_k}\mathcal{S}^{d-1}$ is a tangent-space gradient, and $\eta > 0$ is a learning rate. In practice, we further scaled $g$ by its norm. This was inspired by Euclidean space optimizers (Hinton et al., 2012; Kingma & Ba, 2015; Loshchilov & Hutter, 2019), which normalize gradients based on moving averages of squared gradients. See Algorithm 1 and Appendix C.1 for further details.

### 3.2    MAXIMUM A POSTERIORI FORMULATION WITH A DIRECTIONAL vMF PRIOR

To incorporate a directional prior, we formulate the optimization for the optimal direction $v^*$ as a Maximum A Posteriori (MAP) estimation problem. Given a dataset of images $\mathcal{D} = \{z_1, \ldots, z_n\}$, the MAP estimate is found by maximizing the posterior probability:

$$v^* = \arg\max_{\mathbf{v}} p(\mathbf{v} \mid \mathcal{D}) = \arg\max_{\mathbf{v}} \left[\log p(\mathcal{D} \mid \mathbf{v}) + \log p(\mathbf{v})\right]. \qquad (3)$$

Minimizing the negative log-posterior is equivalent to minimizing a loss function composed of a data term and a prior term, $\mathcal{L}(v) = \mathcal{L}_{\text{data}}(m^* v) + \mathcal{L}_{\text{prior}}(v)$.

The data term, $\mathcal{L}_{\text{data}} = -\log p(\mathcal{D} \mid \mathbf{v})$, is the negative log-likelihood of the images given the direction. Following standard practice for diffusion models (Ho et al., 2020), we use the mean squared error (MSE) between the true and predicted noise as the objective:

$$\mathcal{L}_{\text{data}}(m^* v) := \mathbb{E}_{z,t,\epsilon}[\|\epsilon - \epsilon_\theta(z_t, t, c(m^* v))\|_2^2]. \qquad (4)$$

Here, $\epsilon_\theta$ and $c(\cdot)$ are the diffusion model and text encoder, respectively. For notational simplicity, we write $c(m^* v)$ as shorthand for the text conditioning obtained from a sampled prompt template containing the personalized token with embedding $m^* v$, and omit the explicit dependence on text prompt throughout the paper.

For the prior term, $-\log p(\mathbf{v})$, we use a von Mises-Fisher (vMF) distribution on the direction $\boldsymbol{v}$ (detailed justification in Appendix C.2). The vMF distribution is a probability distribution on the $(d-1)$-sphere, analogous to the Gaussian distribution in Euclidean space. It is parameterized by a mean direction $\boldsymbol{\mu} \in \mathcal{S}^{d-1}$ and a concentration parameter $\kappa \geq 0$. The probability density function is given by:

$$p(\mathbf{v}|\boldsymbol{\mu}, \kappa) = \frac{\kappa^{d/2-1}}{(2\pi)^{d/2} I_{d/2-1}(\kappa)} \exp(\kappa \boldsymbol{\mu}^\mathsf{T} \boldsymbol{v}), \tag{5}$$

where $I_{d/2-1}$ is the modified Bessel function of the first kind. Here, we work with unnormalized density: $p(\mathbf{v}) \propto \exp(\kappa \boldsymbol{\mu}^\mathsf{T} \boldsymbol{v})$. Ignoring constants, the negative log-prior yields our regularization term, $\mathcal{L}_{\mathrm{prior}}(\boldsymbol{v}) = -\kappa \boldsymbol{\mu}^\mathsf{T} \boldsymbol{v}$.

**Constant-direction prior gradient.** A useful property is that the Euclidean gradient of the regularization term is a constant: $\nabla_{\boldsymbol{v}}(-\kappa \boldsymbol{\mu}^\mathsf{T} \boldsymbol{v}) = -\kappa \boldsymbol{\mu}$. Practically, we just add this vector to the data gradient before projecting to the tangent space and retracting. This is analogous in spirit to decoupled weight decay (Loshchilov & Hutter, 2019), but adapted for the sphere with a directional prior. The update is computationally cheap (requiring no new graph operations), numerically stable, and highly interpretable: it applies a *constant pull* towards a semantically meaningful direction.

**Selection of vMF parameters.** The vMF prior is defined by a mean direction $\boldsymbol{\mu}$ and a concentration parameter $\kappa$. The mean direction $\boldsymbol{\mu}$ is set to the normalized embedding of a corresponding class token (e.g., 'dog') from the pre-trained text encoder and is held constant during optimization. Since estimating $\kappa$ is non-trivial, we treat it as a hyperparameter that controls the strength of the prior. We performed a grid search and found that values in the range of `5e-5` to `2e-4` work well. Based on this, we simply fixed the value of $\kappa$ to `1e-4` for all experiments. Further discussion on the selection of prior can be found in Appendix D.2 and D.4.

## 4 EXPERIMENTS

### 4.1 EXPERIMENTAL SETUPS

All experiments were implemented using PyTorch (Paszke et al., 2019) and the HuggingFace `diffusers` library (von Platen et al., 2022), with a single NVIDIA A6000 GPU. Detailed implementation specifications are provided in Appendix D.1.

**Datasets.** For subject personalization, we employed all reference images from the DreamBooth dataset (Ruiz et al., 2023). Additional experiments on stylization and face personalization are presented in Appendix D.7, utilizing StyleDrop (Sohn et al., 2023) and images from FFHQ (Karras et al., 2019). We evaluated all methods using 40 prompts, comprising the complete set of prompts from the DreamBooth dataset supplemented with 10 additional complex prompts.

**Models.** Unless otherwise specified, we employed Stable Diffusion XL (SDXL) (Podell et al., 2024) as our primary model due to its superior performance and widespread adoption in recent literature. To demonstrate DTI's applicability to more recent architectures, we conducted additional experiments on SANA 1.5 (Xie et al., 2025), which employs Gemma (Team et al., 2024) as the text encoder and DiT (Peebles & Xie, 2023) as the image generator.

**Baselines.** Our method extends Textual Inversion (TI) (Gal et al., 2023a), serving as our primary baseline for direct comparison. We additionally evaluate against CrossInit (Pang et al., 2024a), an enhanced TI variant that incorporates specialized initialization and regularization techniques. Comprehensive comparisons with additional baselines, including P+ (Voynov et al., 2023), NeTI (Alaluf et al., 2023), CoRe (Wu et al., 2025), and DCO (Lee et al., 2024) are provided in Appendix D.3.

**Metrics.** Following established evaluation protocols (Ruiz et al., 2023; Kumari et al., 2023; Gal et al., 2023a), we assessed each method across two primary dimensions: subject fidelity and image-text alignment. Subject fidelity was quantified using DINOv2 (Oquab et al., 2023) feature cosine similarity. For image-text alignment, we employed SigLIP (Zhai et al., 2023), a more recent variant of CLIP, following recent work (Lee et al., 2024). For each instance, we generated samples from 40 text prompts using 4 random seeds, yielding 160 samples per instance. Complete evaluation details are provided in Appendix D.1. Results were further validated through a user study conducted via Amazon Mechanical Turk.

Table 2: Our DTI consistently improves baselines by generating outputs with enhanced text fidelity while maintaining subject similarity.

| Methods | SDXL | | SANA 1.5-1.6B | | SANA 1.5-4.8B | |
|---|---|---|---|---|---|---|
| | Image | Text | Image | Text | Image | Text |
| TI | **0.561** | 0.292 | **0.480** | 0.621 | 0.446 | 0.646 |
| TI-rescaled | 0.243 | 0.466 | 0.253 | 0.655 | 0.287 | 0.548 |
| CrossInit | 0.545 | 0.464 | 0.344 | 0.614 | 0.299 | 0.622 |
| **DTI (ours)** | 0.450 | **0.522** | 0.479 | **0.744** | **0.452** | **0.757** |

Table 3: Ablation studies. We tested and confirmed the effectiveness of every component of our DTI.

| Optimizer | $m^\star$ | $\kappa \times 10^{-3}$ | Image | Text |
|---|---|---|---|---|
| AdamW | mean | 0.1 | 0.335 | 0.463 |
| RSGD | min | 0.1 | 0.030 | 0.074 |
| RSGD | 5.0 (OOD) | 0.1 | 0.383 | 0.373 |
| RSGD | mean | 0.0 | **0.507** | 0.436 |
| RSGD | mean | 0.5 | 0.278 | **0.688** |
| RSGD | mean | 0.1 | 0.450 | 0.522 |

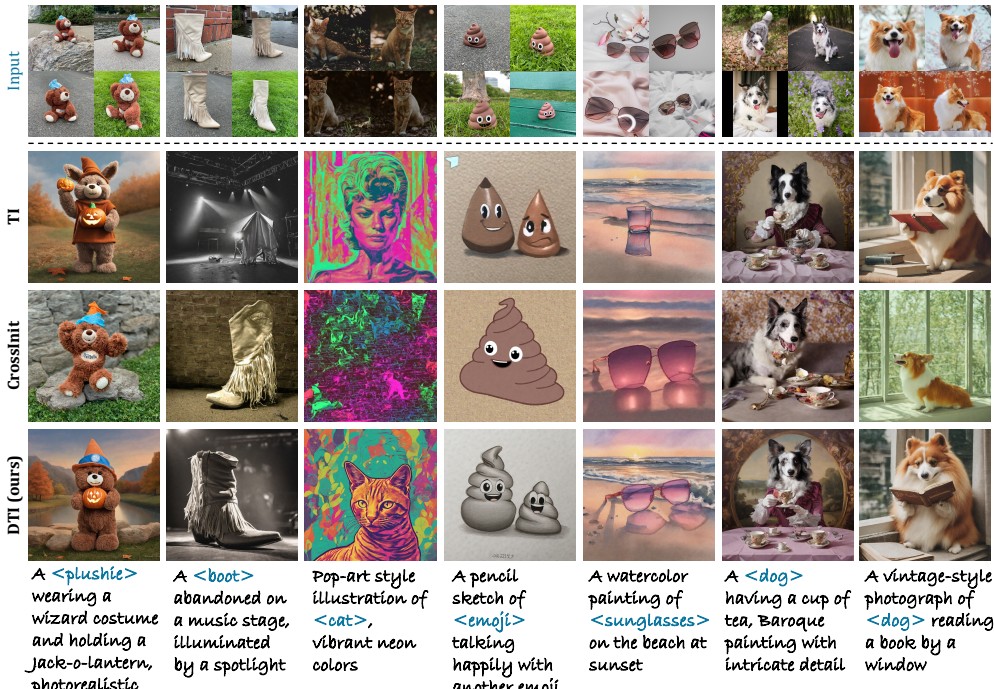

A <plushie> wearing a wizard costume and holding a Jack-o-lantern, photorealistic digital art

A <boot> abandoned on a music stage, illuminated by a spotlight

Pop-art style illustration of <cat>, vibrant neon colors

A pencil sketch of <emoji> talking happily with another emoji

A watercolor painting of <sunglasses> on the beach at sunset

A <dog> having a cup of tea, Baroque painting with intricate detail

A vintage-style photograph of <dog> reading a book by a window

Figure 3: We compare DTI with previous methods across diverse subjects and textual prompts, spanning simple descriptions to complex variations in attributes, backgrounds, and styles (same random seeds). All results in this figure are generated with SDXL (SANA in Appendix Figure 9).

## 4.2 MAIN RESULTS

**Quantitative results.** In Table 2, we quantitatively evaluate DTI along two axes: subject similarity and text–prompt fidelity. DTI consistently produces outputs that adhere closely to the prompt while maintaining high subject similarity. To isolate the role of embedding norm analyzed in Section 2.2, we rescaled TI's learned embeddings to the in-distribution norm—specifically, the average norm of the vocabulary embeddings, matching the norm scale used in DTI. Consistent with our analysis, this simple rescaling noticeably improves text fidelity but does not fully resolve the problem, as it degrades image similarity. CrossInit achieves strong text fidelity on SDXL but fails to do so consistently on SANA, which we attribute to differences in their text encoders; SDXL uses a CLIP text encoder, while SANA employs the LLM-based encoder. Notably, DTI's advantage over the baselines becomes even more pronounced as the model size increases. Overall, these results clearly demonstrate the advantage of DTI over competing baselines. Additional comparisons with further baselines on other Stable Diffusion variants are provided in Appendix D.3.

**Qualitative results.** Figure 3 illustrates qualitative comparisons across various prompts. DTI consistently generates images that more accurately reflect the content of the captions, while effectively preserving subject consistency. For instance, for 'Pop-art style illustration of <cat>', TI omits the

cat while DTI renders the cat in the specified style. Similarly, in the second column, TI and CrossInit fail to incorporate all elements of the prompt, disregarding either the subject or details such as 'music stage' and 'spotlight'. In contrast, DTI integrates both the subject and these details, producing a more complete output. Collectively, these examples highlight DTI's superior compositional fidelity and subject preservation, demonstrating that it consistently satisfies all prompt constraints. We attribute this to DTI's stable optimization within the directional space, which facilitates improved integration of multiple prompt components. DTI's ability to maintain subject fidelity and adhere to textual intent establishes it as a robust choice for a wide range of text-to-image generation tasks. Additional qualitative results including those of SANA can be found in Appendix D.6.

### 4.3 ABLATION STUDY

We performed an ablation study to verify the effectiveness of components of our DTI, including the optimization space, the embedding magnitude $m$, and the concentration parameter $\kappa$ of the vMF distribution. The results are summarized in Table 3. To validate our choice of Riemannian SGD (RSGD), we compared it against a baseline using the AdamW optimizer. This baseline performs standard Euclidean updates and then projects the vector back onto the unit sphere after each step, which is not a true Riemannian update. The results show that RSGD substantially outperforms AdamW, highlighting the benefit of respecting the geometry of the directional manifold. Next, we found that fixing the magnitude to the minimum or to an out-of-distribution scale negatively affected either subject similarity or text fidelity. Setting the magnitude to an in-distribution scale yields the best results. Lastly, removing the prior (i.e., $\kappa = 0$) or extremely high values of $\kappa$ hurts the performance, while moderate incorporation of the prior provides the most stable results. Overall, we confirm that these ablation results validate our design choices. Further analyses are provided in Appendix D.4.

### 4.4 HUMAN EVALUATION

To further examine the effectiveness of our method, we conducted a large-scale user study (100 participants via *Amazon Mechanical Turk*) to measure real-world user preferences. Each participant was asked to respond to 20 questions, comprising 10 questions assessing subject fidelity and 10 questions evaluating image-text alignment. Participants were instructed to select the output that best met the specified criteria for each question. To ensure the reliability of the study, we excluded four user responses that did not adhere to the specified instructions.

Table 4: We surveyed real-world user preferences regarding subject fidelity and image-text alignment. DTI ranks the top in both metrics, confirming its practical benefits.

|  | TI | CrossInit | **DTI (ours)** |
|---|---|---|---|
| Image fidelity | 13.78 | 42.87 | **43.45** |
| Text alignment | 10.83 | 22.40 | **66.77** |

A fixed random seed was employed, and the answer options were shuffled for each question. The results, summarized in Table 4, show that DTI consistently outperforms the other methods on both metrics, indicating that its improvements in alignment are clearly perceived by human evaluators. More details of this user study can be found in Appendix D.5.

### 4.5 EMBEDDING INTERPOLATION FOR CREATIVE APPLICATIONS

We demonstrate the creative potential of our DTI through embedding interpolation experiments. As illustrated in Figure 4, our DTI generates coherent interpolations via spherical linear interpolation (SLERP), which matches the unit-sphere parameterization. This capability is a direct result of DTI's unit-spherical embedding space, which enables smooth and effective transitions. In contrast, the linear interpolation used by TI often fails to produce coherent intermediate results.

The advantages of our approach are clearly visible across different domains. As shown in the first rows of the figure, one can seamlessly merge a dog and a teapot, resulting in imaginative hybrid objects like an adorable teapot that progressively adopts the features of the dog. This indicates that DTI excels at blending conceptually distinct subjects, a significant creative application. In the second example, it can create a creative animal between a dog and a cat, that merges the features of each animal in a smooth manner. Lastly, DTI smoothly interpolates between the faces of a young boy and an older woman, generating a plausible progression that simultaneously alters age and appearance while maintaining facial coherence. This highlights its potential for nuanced face personalization.

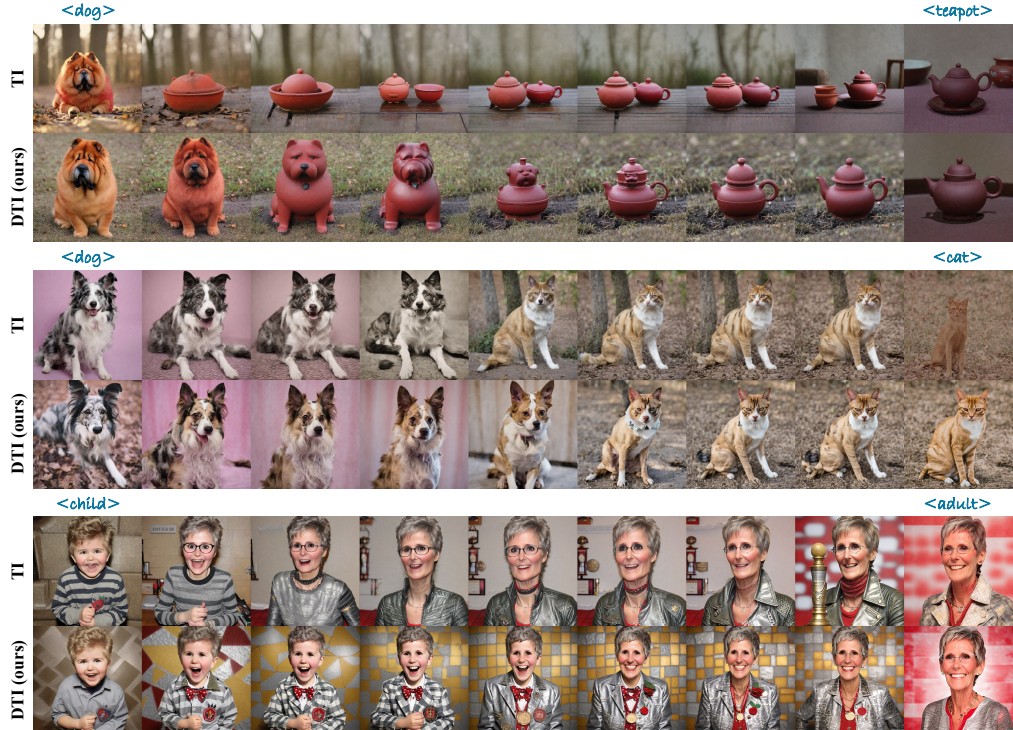

Figure 4: We compare images generated by a TI and our DTI. Two personalized subjects are interpolated, including interpolation between inanimate and animate subjects, live subjects, and human faces. Images are generated with interpolation ratio $[0.0, 0.35, 0.40, 0.45, 0.50, 0.55, 0.60, 0.65, 1.0]$ for better visualization. Our DTI offers smooth interpolation between concepts, expanding personalization along a more creative axis.

Throughout these transitions, DTI produces visually consistent and creative outputs that retain semantic meaning, unlocking novel user-driven applications and establishing it as a powerful tool for intuitive concept blending. We provide the results of other applications, including face personalization, stylization and subject-style generation throughout Appendix D.7.

## 5 RELATED WORK

### 5.1 PERSONALIZED TEXT-TO-IMAGE GENERATION

Recent advancements in text-to-image (T2I) generation have considerably expanded the creative capabilities and flexibility of generative models (Ramesh et al., 2021; Rombach et al., 2022; Nichol et al., 2022; Ramesh et al., 2022; Yu et al., 2022; Podell et al., 2024). Despite these innovations, natural language inherently struggles to precisely convey nuanced, user-specific concepts. This inherent limitation has driven the development of personalization methods, which allow users to generate images reflecting unique concepts with creative prompts.

Textual Inversion (Gal et al., 2023a), which is most well-known for its lightweight integration into many other personalization works, uses embedding optimization by introducing learnable tokens for personalized information without model modification. Subsequent work explored diverse embedding strategies (Voynov et al., 2023; Alaluf et al., 2023; Wu et al., 2025; Zhang et al., 2024a), often at excessive computational cost. Among them, CrossInit (Pang et al., 2024a) offered an efficient initialization strategy with minimal overhead, replacing initialization tokens with the output of text encoder and using a regularization loss.

In contrast, fine-tuning based methods such as DreamBooth (Ruiz et al., 2023) achieve high subject fidelity, but require significant computational resources compared to embedding optimization meth-

ods (Kumari et al., 2023; Han et al., 2023; Gu et al., 2023; Chen et al., 2023a; Tewel et al., 2023; Zhang et al., 2024b; Qiu et al., 2023; Pang et al., 2024b) . More recently, Park et al. (2024) proposed fine-tuning the text encoder instead of the image generator for efficiency, but they still demand more parameters compared to embedding optimization methods.

Meanwhile, there exists a line of encoder-based approaches (Wei et al., 2023; Ruiz et al., 2024; Ye et al., 2023; Gal et al., 2023b; Chen et al., 2023b; Li et al., 2023; Pang et al., 2024b; Ma et al., 2024) that offer fast inference, but they necessitate substantial pre-training.

## 5.2 DIRECTIONAL EMBEDDING SPACE

A number of prior works have emphasized constraining embedding representations to the hypersphere. These include using vMF mixtures for directional clustering (Jameel & Schockaert, 2019), normalizing norms for face recognition (Meng et al., 2019), angle-optimized embeddings to address cosine saturation (Li & Li, 2024), and spherical constraints for uniform document clustering (Zhang et al., 2020). Wang & Isola (2020) offered theoretical support for hyperspherical constraints in contrastive learning. Our method aligns with this trend by modeling embeddings as directional distributions but uniquely decomposes and explicitly optimizes textual embedding direction using a vMF prior within the Textual Inversion framework.

## 6 DISCUSSION & CONCLUSION

Our DTI primarily improves text prompt fidelity as it does not directly optimize for subject similarity. For applications where high subject fidelity is paramount, DTI can be used in conjunction with complementary lightweight fine-tuning methods, such as LoRA, as we demonstrate qualitatively in Figure 11. Furthermore, our analysis is centered on the geometry of modern pre-norm text encoders. An interesting direction for future work would be to investigate whether our findings generalize to other types of encoders with different normalization or positional encoding schemes.

Overall, our work tackles a key challenge in personalized text-to-image generation: achieving a strong alignment between text prompts and generated imagery. We have identified and rigorously analyzed embedding norm inflation as a significant bottleneck to this alignment, providing both theoretical and empirical evidence of its detrimental effects. In addition, our investigation focuses on the directional characteristics of the token embedding space, an area that has been comparatively underexplored in the literature, particularly when contrasted with the extensive research dedicated to the output embedding space of text encoders. Leveraging this key insight into the semantic significance of token embedding directionality, we proposed Directional Textual Inversion (DTI), a novel framework that keeps the embedding norm at an in-distribution scale and solely optimizes the direction. We further reformulate the conventional Textual Inversion optimization process by incorporating directional priors. Our DTI demonstrably enhances prompt fidelity, thereby substantially improving the practicality of token embedding-based personalization and enabling innovative creative applications such as the smooth interpolation of learned concepts. We truly hope our work paves the way for more effective and versatile token embedding-based personalization within generative AI, unlocking enhanced capabilities for users to articulate their unique creative visions with greater precision and control.

ACKNOWLEDGMENTS

This research was supported by the Basic Science Research Program through the National Research Foundation of Korea (NRF) funded by the MSIP (No. RS-2025-00520207); the Institute of Information & Communications Technology Planning & Evaluation (IITP) grants funded by the Korea government (MSIT) (Nos. 2022-0-00680, 2022-0-01045, RS-2024-00457882, RS-2025-02217259, RS-2019-II190075), including the National AI Research Lab Project and the Artificial Intelligence Graduate School Support Program (KAIST); and the Korea Evaluation Institute of Industrial Technology (KEIT) grants funded by the Korea government (MOTIE) (Nos. 2022-0-00680, 2022-0-01045, RS-2025-02217259).

REPRODUCIBILITY STATEMENT

To ensure the full reproducibility of our research, we provide our complete source code, experimental details, and dataset information. We utilized publicly available datasets, primarily from Dream-Booth (Ruiz et al., 2023), FFHQ (Karras et al., 2019), and StyleDrop (Sohn et al., 2023), and our repository includes scripts for all necessary preprocessing. Additionally, all software dependencies are explicitly specified in the `pyproject.toml` file. All experiments were conducted on a single NVIDIA A6000 GPU, with training taking approximately 7 minutes per subject for SDXL-base and 30 minutes for SANA 1.5-1.6B. To guarantee transparency and ease of use, all hyperparameters are detailed in the Appendix and are also included in the run scripts within our codebase.

LLM USAGE STATEMENT

We utilized Large Language Models (LLMs) to improve the grammar and clarity of this manuscript. The core research, including the analysis and method, is the exclusive work of the authors.

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

## A EMBEDDING NORM AND DIRECTION

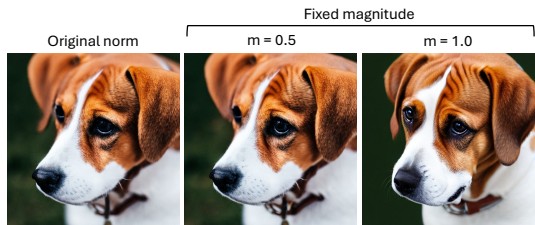

Figure 5: **Effect of magnitude change.** We set the magnitude to a fixed value to analyze the impact of magnitude changes. The resulting outputs show no noticeable difference.

We altered the magnitude of the token as exemplified in Figure 5. However, the resulting output remained mostly unchanged. This indicates that minor adjustments to the magnitude do not significantly affect the outcome.

Table 5: **Nearest tokens under different measures.** We show the nearest tokens to the query words 'study' and 'writing' using both cosine similarity and Euclidean distance.

| Query | Cosine | Euclidean |
|---|---|---|
| **study** | studies, studying, research, bookclub, reading, studied, sketches, measurements, thumbnail | U+3160, texanscheer, asober, instaweatherpro, mydayin, premiosmtvmiaw, tairp, thepersonalnetwork, U+2412 |
| **writing** | writer, write, written, writ, writers, writings, recording, blogging, wrote | phdlife, poetryday, tomorrowspaper, urstrulymahesh, @___, twitterkurds, asober, fakespeare, jamiedor |

In Table 5, we provide additional examples illustrating the nearest words retrieved for each query under different similarity measures, which strongly correlate with either direction or magnitude. Our analysis reveals that cosine similarity retrieves words that share semantic meaning with the query. Conversely, Euclidean distance is significantly affected by embedding magnitude, often retrieving words with limited or no semantic relevance. This demonstrates that semantic meaning is predominantly associated with embedding direction rather than magnitude. Note that words beginning with U+ denote Unicode.

## B PROOFS FOR THEORETICAL STATEMENTS

### B.1 SETUP

**Pre-norm block.** We study *pre-norm* Transformer blocks

$$\boldsymbol{x}^{(\ell+1)} = \boldsymbol{x}^{(\ell)} + F_\ell(\text{Norm}(\boldsymbol{x}^{(\ell)})), \quad \ell = 0, \dots, L-1, \tag{6}$$

where $\text{Norm} \in \{\text{LayerNorm}, \text{RMSNorm}\}$ (with optional affine $(\gamma, \beta)$ absorbed into $F_\ell$).

**Scale invariance.** For normalizations, we use the standard, scale-invariant definitions:

$$\text{RMSN}(\boldsymbol{x}) = \sqrt{d}\frac{\boldsymbol{x}}{\|\boldsymbol{x}\|_2}, \quad \text{LN}(\boldsymbol{x}) = \sqrt{d}\frac{\boldsymbol{C}\boldsymbol{x}}{\|\boldsymbol{C}\boldsymbol{x}\|_2}, \quad \boldsymbol{C} := \boldsymbol{I} - \frac{1}{d}\boldsymbol{1}\boldsymbol{1}^\top. \tag{7}$$

Thus $\text{RMSN}(s\boldsymbol{x}) = \text{RMSN}(\boldsymbol{x})$ and $\text{LN}(s\boldsymbol{x}) = \text{LN}(\boldsymbol{x})$ for all $s > 0$. Please refer to original papers (Ba et al., 2016; Zhang & Sennrich, 2019) for further details.

**Token decomposition.** For the input token, we denote $\boldsymbol{x}^{(0)} = m\boldsymbol{v} + \boldsymbol{p}$ with $m > 0$, $\|\boldsymbol{v}\|_2 = 1$, and (optional) absolute positional embedding $\boldsymbol{p} \in \mathbb{R}^d$.

**Bounded sub-layers.** Define $\mathcal{S} = \{\text{Norm}(\boldsymbol{z}) : \boldsymbol{z} \neq \boldsymbol{0}\}$. Since Norm maps into a fixed scale, bounded set and $F_\ell$ (attention/MLP plus projections) is continuous on bounded sets,

$$B_\ell := \sup_{\boldsymbol{u} \in \mathcal{S}} \|F_\ell(\boldsymbol{u})\|_2 < \infty. \tag{8}$$

Throughout, $\|\cdot\|_2$ denotes the Euclidean ($l_2$) norm.

## B.2 POSITIONAL ATTENUATION

**Lemma 1** (Absolute positional embedding attenuates as $m \to \infty$). *Let $\boldsymbol{x}^{(0)} = m\boldsymbol{v} + \boldsymbol{p}$ with $\|\boldsymbol{v}\|_2 = 1$, $m > 0$, and absolute positional embedding $\boldsymbol{p} \in \mathbb{R}^d$. Suppose Norm $\in \{LayerNorm, RMSNorm\}$ and $\boldsymbol{v}$ is non-degenerate for LayerNorm (i.e., its per-feature variance is nonzero; this holds for generic token embeddings). Then*

$$\left\|\text{Norm}(m\boldsymbol{v} + \boldsymbol{p}) - \text{Norm}(m\boldsymbol{v})\right\|_2 = \mathcal{O}(\frac{\|\boldsymbol{p}\|_2}{m}) \quad \text{as } m \to \infty \text{ (with } \boldsymbol{v}, \boldsymbol{p} \text{ fixed)}.$$

*Hence the positional contribution shrinks linearly in $1/m$.*

*Proof.* By scale invariance, $\text{Norm}(m\boldsymbol{v} + \boldsymbol{p}) = \text{Norm}(\boldsymbol{v} + \varepsilon)$ with $\varepsilon := \boldsymbol{p}/m$, and $\text{Norm}(m\boldsymbol{v}) = \text{Norm}(\boldsymbol{v})$.

*RMSNorm.* With $\|\boldsymbol{v}\| = 1$,

$$\frac{\boldsymbol{v} + \varepsilon}{\|\boldsymbol{v} + \varepsilon\|} = \boldsymbol{v} + (\boldsymbol{I}_d - \boldsymbol{v}\boldsymbol{v}^\top)\varepsilon + \mathcal{O}(\|\varepsilon\|^2),$$

hence $\text{RMSN}(\boldsymbol{v} + \varepsilon) - \text{RMSN}(\boldsymbol{v}) = \sqrt{d}\,(\boldsymbol{I}_d - \boldsymbol{v}\boldsymbol{v}^\top)\varepsilon + \mathcal{O}(\|\varepsilon\|^2)$ and $\|\text{RMSN}(m\boldsymbol{v} + \boldsymbol{p}) - \text{RMSN}(m\boldsymbol{v})\| \leq \sqrt{d}\,\|\boldsymbol{p}\|/m + O(m^{-2})$.

*LayerNorm.* Write $\boldsymbol{a} := \boldsymbol{C}\boldsymbol{v} \neq \boldsymbol{0}$, $\boldsymbol{u} := \boldsymbol{a}/\|\boldsymbol{a}\|$. Then

$$\frac{\boldsymbol{a} + \boldsymbol{C}\varepsilon}{\|\boldsymbol{a} + \boldsymbol{C}\varepsilon\|} = \boldsymbol{u} + \frac{(\boldsymbol{I}_d - \boldsymbol{u}\boldsymbol{u}^\top)\boldsymbol{C}\varepsilon}{\|\boldsymbol{a}\|} + \mathcal{O}(\|\varepsilon\|^2),$$

so $\|\text{LN}(m\boldsymbol{v} + \boldsymbol{p}) - \text{LN}(m\boldsymbol{v})\| \leq \sqrt{d}\,\frac{\|(\boldsymbol{I}_d - \boldsymbol{u}\boldsymbol{u}^\top)\boldsymbol{C}\boldsymbol{p}\|}{m\|\boldsymbol{C}\boldsymbol{v}\|} + \mathcal{O}(m^{-2})$, which is $\mathcal{O}(\|\boldsymbol{p}\|/m)$. $\square$

## B.3 RESIDUAL STAGNATION

**Lemma 2** (Residual stagnation in a pre-norm block). *Let $\boldsymbol{x}^{(\ell+1)} = \boldsymbol{x}^{(\ell)} + F_\ell(\text{Norm}(\boldsymbol{x}^{(\ell)}))$ with $\boldsymbol{x}^{(\ell)} \neq \boldsymbol{0}$ and Norm $\in \{\text{LN}, \text{RMSN}\}$, and let*

$$B_\ell := \sup_{\boldsymbol{u} \in \mathcal{S}} \|F_\ell(\boldsymbol{u})\|_2 < \infty.$$

*Then*

$$\frac{\|\boldsymbol{x}^{(\ell+1)} - \boldsymbol{x}^{(\ell)}\|_2}{\|\boldsymbol{x}^{(\ell)}\|_2} \leq \frac{B_\ell}{\|\boldsymbol{x}^{(\ell)}\|_2}.$$

*If additionally $B_\ell < \|\boldsymbol{x}^{(\ell)}\|_2$,*

$$\angle(\boldsymbol{x}^{(\ell)}, \boldsymbol{x}^{(\ell+1)}) \leq \arcsin\Big(\frac{B_\ell}{\|\boldsymbol{x}^{(\ell)}\|_2}\Big).$$

*Proof.* Since $\text{Norm}(\boldsymbol{x}^{(\ell)}) \in S$, we have $\|\boldsymbol{x}^{(\ell+1)} - \boldsymbol{x}^{(\ell)}\|_2 = \|F_\ell(\text{Norm}(\boldsymbol{x}^{(\ell)}))\|_2 \leq B_\ell$, giving the first bound. Write $\boldsymbol{x}^{(\ell+1)} = \boldsymbol{x}^{(\ell)} + \delta$. The orthogonal component of $\delta$ is at most $\|\delta\|$; a short calculation shows $\sin\angle(\boldsymbol{x}^{(\ell)}, \boldsymbol{x}^{(\ell+1)}) \leq \|\delta\|_2/\|\boldsymbol{x}^{(\ell)}\|_2 \leq B_\ell/\|\boldsymbol{x}^{(\ell)}\|_2$, which implies the stated angle bound. $\square$

**Proposition 1** (Accumulated directional drift across $L$ pre-norm blocks). *Let $\boldsymbol{x}^{(0)} \neq \boldsymbol{0}$ and $\boldsymbol{x}^{(\ell+1)} = \boldsymbol{x}^{(\ell)} + F_\ell(\text{Norm}(\boldsymbol{x}^{(\ell)}))$ for $\ell = 0, \ldots, L-1$. Let $B_\ell := \sup_{\boldsymbol{u} \in S} \|F_\ell(\boldsymbol{u})\|_2 < \infty$, and $S_L := \sum_{j=0}^{L-1} B_j$. Assume $\|\boldsymbol{x}^{(0)}\|_2 > S_L$, then*

$$\angle(\boldsymbol{x}^{(0)}, \boldsymbol{x}^{(L)}) \leq \frac{\pi}{2} \sum_{\ell=0}^{L-1} \frac{B_\ell}{\|\boldsymbol{x}^{(0)}\|_2 - \sum_{j<\ell} B_j} \leq \frac{\pi}{2} \frac{S_L}{\|\boldsymbol{x}^{(0)}\|_2 - S_L}.$$

*Proof.* Let $\theta_\ell := \angle(x^{(\ell)}, x^{(\ell+1)})$. By Lemma 2 above, $\theta_\ell \leq \arcsin\big(B_\ell/\|x^{(\ell)}\|\big) \leq \frac{\pi}{2} B_\ell/\|x^{(\ell)}\|$. Also $\|x^{(\ell)}\| \geq \|x^{(0)}\| - \sum_{j<\ell} B_j$ (each step $j$ can shrink the norm by at most $B_j$). Summing angles (spherical triangle inequality) gives the first display; since $\|x^{(0)}\| - \sum_{j<\ell} B_j \geq \|x^{(0)}\| - S_L$, each fraction is $\leq B_\ell/(\|x^{(0)}\| - S_L)$, yielding the last bound. $\square$

## C  EXTENDED METHODS

### C.1  RSGD FOR TOKEN EMBEDDING OPTIMIZATION

We observe that gradient magnitudes tend to increase as training progresses, which often leads to instability in the later stages. Although standard learning rate schedules can help mitigate this issue, the gradient dynamics vary considerably across different datasets and training settings, limiting the effectiveness of fixed schedules. To address this, we draw inspiration from adaptive optimization techniques in Euclidean space (Kingma & Ba, 2015; Duchi et al., 2011) and propose a simple yet effective gradient scaling scheme based on gradient norms:

$$\boldsymbol{g}'_k = \boldsymbol{g}_k / \|\boldsymbol{g}_k\|_2, \tag{9}$$

where $\boldsymbol{g}_k$ is the gradient at iteration $k$. This normalization is equivalent to using an adaptive step size $\eta/\|\boldsymbol{g}_k\|$ in a Euclidean update $\boldsymbol{v}_{k+1} = \boldsymbol{v}_k - (\eta/\|\boldsymbol{g}_k\|_2)\boldsymbol{g}_k$; the update direction is still $\boldsymbol{g}_k$, but the length is fixed to $\eta$, preventing very large gradients from causing excessively large parameter updates. Note that a similar technique was previously explored in the context of Riemannian optimization (Cho & Lee, 2017).

### C.2  WHY vMF OVER OTHER DISTRIBUTIONS?

We chose the von Mises-Fisher (vMF) distribution as it is ideally suited for modeling the directional characteristics of token embeddings we identified in Section 2. Our central hypothesis is that the token embedding vocabulary can be modeled as a **mixture of vMF distributions**, where each component corresponds to a distinct semantic cluster (e.g., one for animals, another for objects). The vMF distribution is a suitable building block for this model for three key reasons:

- **It's a natural fit.** The vMF is the natural analog to the Gaussian distribution on a hypersphere, making it a principled and standard choice for modeling directional data clusters.

- **It's computationally efficient.** The vMF's mathematical form is exceptionally convenient for optimization. In our MAP formulation, the gradient of the log-prior is a *constant-direction vector* ($-\kappa\mu$), which provides a stable and efficient semantic pull without requiring complex calculations. This simplicity makes it more suitable for high-dimensional embeddings in large-scale models than alternatives like the Kent and Bingham distributions.

- **It's interpretable and controllable.** The parameters are easy to understand. The mean direction $\boldsymbol{\mu}$ serves as a *semantic anchor* to prevent the learned token from drifting away from related concepts, while the concentration $\kappa$ allows us to control the strength of this regularization.

These factors collectively make the vMF distribution a superior choice for our application, providing the necessary regularization in a way that is both mathematically principled and computationally tractable.

## D  EXTENDED EXPERIMENTS

### D.1  IMPLEMENTATION DETAILS

Following the protocol of recent studies, we primarily conducted experiments using Stable Diffusion XL (SDXL). To demonstrate broader applicability to different models, we also conducted experiments with the recent model SANA 1.5 (Xie et al., 2025); these results are presented in Table 2.

For a fair comparison, we adopted most hyperparameter settings from the Textual Inversion (TI) implementation provided by the Hugging Face `diffusers` library (von Platen et al., 2022). We used a training batch size of 4 and enabled mixed-precision training with the bfloat16 (bf16) format. For the baselines, the learning rate was set to the commonly used value of $5 \times 10^{-3}$. Since our method employs a different optimizer, we conducted a separate learning rate search and set it to 0.02. Following Rombach et al. (2022) and Gal et al. (2023a), we scaled the base learning rate proportionally to the batch size for all methods.

All experiments were performed with a fixed random seed of 42, and the maximum number of training steps was set to 500 for SDXL and 1000 for SANA. For image generation, we used the `DDIMScheduler` with 50 inference steps for SDXL and the `FlowMatchEulerDiscreteScheduler` with 20 inference steps for SANA.

**Hyperparameters.** There are various approaches to selecting the concentration parameter $\kappa$. We performed a grid search and found that values in the range of $5 \times 10^{-5}$ to $2 \times 10^{-4}$ work well. Therefore, we did not conduct an exhaustive search for a more precise optimal value. Throughout our experiments, we simply fixed $\kappa$ at $10^{-4}$, which generalizes well across different settings. Examples illustrating the effects of varying $\kappa$ are provided in Table 3.

**Baselines.** Throughout this paper, we compare our method with two baseline approaches: Textual Inversion (TI) (Gal et al., 2023a) and CrossInit (Pang et al., 2024a). Since the official CrossInit implementation is based on Stable Diffusion v2.1 with hyperparameters tailored to that version, we reconfigure it to operate on SDXL by aligning its training setup with that of TI. Specifically, we adopt the same hyperparameters as used for TI, and we set the regularization weight for CrossInit to $10^{-5}$, as specified in the original paper.

## D.2 ON THE CHOICE OF PRIOR

For all of our experiments in the main section, we used the same class tokens as those specified in the DreamBooth dataset as priors. However, we would like to note that since our DTI can leverage the prior, searching for better priors can lead to better results. This demonstrates the effectiveness of the prior.

To test this, we experimented with having a VLM recommend initial tokens. More specifically, we provided reference images to the VLM and asked it to recommend 1-2 words that best describe them. For the experiments, we used Qwen-VL 2.5 (Bai et al., 2025) as the VLM. The results are shown in Table 6.

The results indicate that changing the prior affects performance, although the overall effect is modest. For both TI and our DTI, Qwen-VL initialization tends to increase subject similarity, accompanied by a slight decrease in text fidelity. Practitioners may leverage VLMs or manually craft priors with targeted terms to emphasize desired attributes. Overall, these findings demonstrate the flexibility and effectiveness of leveraging priors.

Table 6: Results with VLM-recommended priors. We compare Qwen-VL recommended initial tokens with DreamBooth initial tokens as priors for DTI.

| Method | Initialization | SDXL | | SANA | |
|---|---|---|---|---|---|
| | | Image | Text | Image | Text |
| TI | DreamBooth init | 0.561 | 0.292 | 0.480 | 0.621 |
| | Qwen-VL init | 0.583 | 0.273 | 0.501 | 0.619 |
| DTI (ours) | DreamBooth init | 0.450 | 0.522 | 0.479 | 0.744 |
| | Qwen-VL init | 0.520 | 0.391 | 0.504 | 0.697 |

## D.3 COMPARISON WITH OTHER BASELINES

We expand our comparative analysis to include additional baselines: P+ (Voynov et al., 2023), NeTI (Alaluf et al., 2023), and CoRe (Wu et al., 2025). We run these experiments mainly on SD1.5 and SD2.1-base as these baseline papers work on those versions. Adhering to the evaluation protocol of the main paper, we measure subject similarity using DINOv2 similarity and prompt fidelity with the CLIP-variant, SigLIP. The results demonstrate that across both architectures, DTI consistently achieves the most favorable balance between these metrics compared to all baselines. Qualitative comparison can be found in Figure 10.

**DTI as a drop-in replacement for TI.** Although DTI is primarily designed for embedding-only personalization, it also functions effectively as a drop-in replacement within fine-tuning pipelines. Recent work on Direct Consistency Optimization (DCO; Lee et al., 2024) typically performs joint

Table 7: **Results on SD1.5 and SD2.1-base.** We compare the baselines that improve TI on different versions of Stable Diffusion. DTI achieves the best balance between subject similarity and text fidelity compared to other baselines.

| Method | SD1.5 | | SD2.1-base | |
|---|---|---|---|---|
| | Image | Text | Image | Text |
| P+ (Voynov et al., 2023) | 0.273 | **0.719** | 0.238 | **0.663** |
| NeTI (Alaluf et al., 2023) | 0.408 | 0.579 | 0.565 | 0.517 |
| CoRe (Wu et al., 2025) | 0.340 | 0.661 | 0.357 | 0.654 |
| DTI (ours) | **0.418** | 0.554 | 0.469 | 0.568 |

optimization of a concept token using standard Textual Inversion (TI). To assess the impact of substituting this TI component with DTI, we conducted joint training for 250 steps using a LoRA module with rank 4.

As shown in Table 8, the conventional TI-based joint training exhibits limited text–alignment (0.456), whereas replacing TI with DTI substantially improves alignment to 0.635. This quantitative improvement is further reflected in Figure 6, where our method accurately incorporates textual attributes (e.g., red backpack), while DCO with TI fails to do so.

Table 8: **DCO experiments.** Comparison of DCO with standard Textual Inversion (TI) versus DCO initialized with our DTI. Integrating our method significantly improves text alignment.

| Method | Image | Text |
|---|---|---|
| DCO | **0.605** | 0.456 |
| DCO + DTI (ours) | 0.568 | **0.635** |

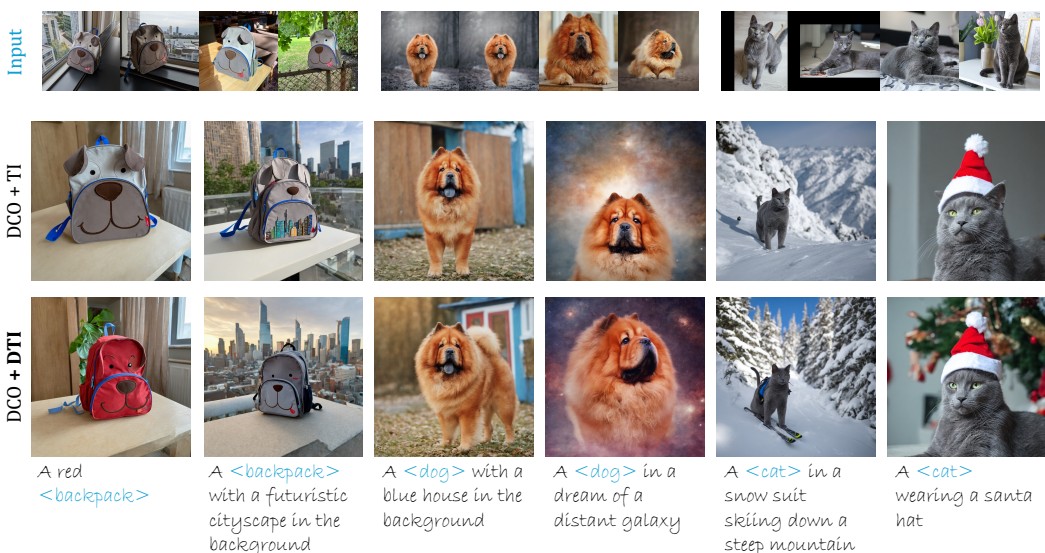

Figure 6: **Qualitative results with DCO (Lee et al., 2024).** We provide a comparison of the output images of DCO + TI and DCO + DTI. The results suggest our DTI's superiority in text fidelity while preserving strong image similarity.

## D.4 ABLATION STUDY

**Effect of Riemannian optimization.** Our DTI framework employs Riemannian optimization to ensure embeddings lie on the spherical manifold $\mathbb{S}^{n-1}$. An alternative is to simply re-scale embeddings after each Euclidean optimization step to achieve this constraint. However, Table 3

(first row) shows this latter Euclidean-based approach with re-scaling achieves suboptimal results, highlighting the benefit of direct Riemannian optimization.

**Effect of magnitude ($m$).** We investigated the impact of the fixed embedding magnitude, $m$, on personalization performance. Our DTI framework, by default, sets $m$ to the average norm observed in the pre-trained CLIP token vocabulary. We compared this "mean" strategy under the Riemannian optimization setting with $\kappa = 1e - 4$:

- Setting $m$ to the minimum vocabulary norm ("min").
- Setting $m$ to the mean vocabulary norm ("mean").
- Setting $m$ to a large, out-of-distribution (OOD) value of 5.0.

As shown in Table 3:

- The "mean" strategy achieves the highest subject similarity and strong text fidelity.
- The "min" strategy results in significantly poorer performance in both metrics.
- Using an OOD magnitude of 5.0 also leads to a degradation in both metrics.

These results validate our choice of fixing the magnitude to an in-distribution scale, specifically the average vocabulary norm, as it provides a strong balance of subject similarity and text alignment. Both excessively small ("min") and out-of-distribution large ("OOD") magnitudes are detrimental.

**Effect of concentration parameter ($\kappa$).** The concentration parameter $\kappa$ of the von Mises-Fisher (vMF) prior controls the strength of the directional regularization. We analyzed its effect by varying $\kappa$ while using Riemannian optimization and the "mean" embedding magnitude. We tested $\kappa = 0.0$ (no prior), $\kappa = 1e - 4$ (DTI default), and $\kappa = 5e - 4$.

The results in Table 3 indicate:

- With $\kappa = 1e - 4$, we observe the best balance between subject similarity and text fidelity.
- Setting $\kappa = 0.0$, which removes the directional prior, leads to lower scores in text fidelity, which suggests that the directional prior improves semantic alignment.
- Increasing the regularization strength with $\kappa = 5e - 4$ yields the highest text fidelity among the tested values but at the cost of reduced subject similarity.

Overall, our default choice of $\kappa = 1e - 4$ provides a better balance between maintaining subject similarity and ensuring text fidelity. Note that $\kappa = 1e - 4$ may not be numerically optimal across all criteria but works reasonably well by providing robust overall performance.

### D.5 DETAILS OF USER STUDY

To evaluate real-world user preferences for image generation quality, we conducted a comprehensive user study involving 100 participants recruited through Amazon Mechanical Turk. Each participant completed a survey consisting of 20 questions, evenly divided into two critical evaluation criteria: subject similarity and text prompt fidelity. For each question, participants were presented with three distinct image options, generated by: Textual Inversion (Gal et al., 2023a), CrossInit (Pang et al., 2024a), and our proposed Directional Textual Inversion (DTI). The order of these three choices was randomized for each question, using a fixed random seed to ensure consistent shuffling across all participants. Sample questions can be found in Figure 7. We collected a total of 96 valid responses, with 4 submissions being excluded due to invalid patterns such as selecting the same answer for all questions. The results, as detailed in Table 4 (in the main paper), demonstrate that our Directional Textual Inversion (DTI) consistently outperforms both Textual Inversion and CrossInit across both evaluation metrics: image subject similarity and text prompt fidelity. These findings confirm the superior performance of our proposed method in generating images that more accurately align with user expectations regarding both visual content and textual descriptions.

### D.6 MORE QUALITATIVE RESULTS

We present additional qualitative comparisons with TI-based approaches (Gal et al., 2023a; Pang et al., 2024a) in Figure 8 (SDXL) and 9 (SANA). The results illustrate that our proposed DTI consistently

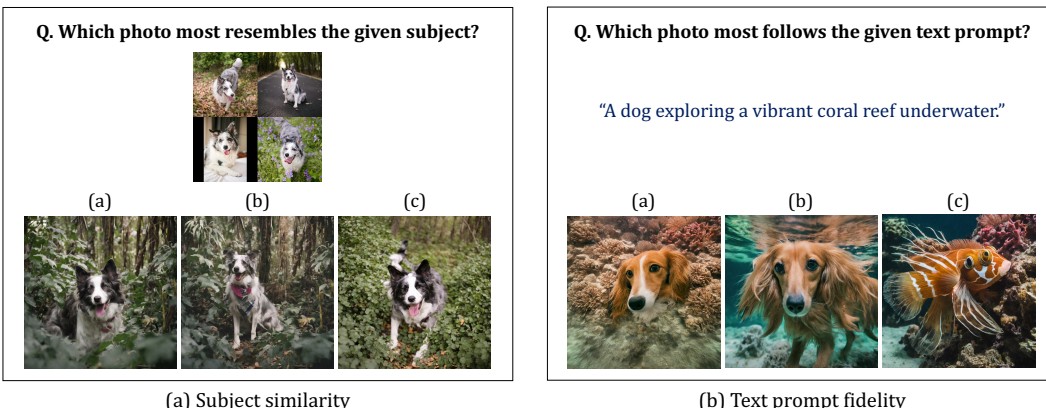

Figure 7: **User study design.** We conducted a user study with 100 participants recruited via Amazon Mechanical Turk to evaluate 20 questions. The evaluation focused on two key aspects: subject similarity (10 questions) and text prompt fidelity (10 questions). To ensure fair comparison, the random seed was fixed and option order was shuffled.

generates outputs that accurately align with the provided text prompts, even in challenging cases where the baseline methods fail to do so.

Our DTI serves as a drop-in replacement for TI, enhancing the model's performance when combined with LoRA. The qualitative results in Figure 11 demonstrate that DTI consistently generates outputs that both precisely follow the text prompt and accurately capture the subject's details.

### D.7    MORE RESULTS ON APPLICATIONS

**Stylization.**    We explore the combination of personalized subject embeddings and style embeddings. Our method, DTI, consistently generates images that accurately reflect both the personalized subject and the specified style. In contrast, TI frequently fails in this task, either by omitting the subject altogether (top row) or by inadequately capturing the intended style or subject details (bottom row) of Figure 12.

**My object in my style.**    We also compare our results in simultaneous generation of personalized subject and style. The results demonstrated in Figure 13 show that DTI successfully generates outputs that are faithful to both subject and style, while TI fails to.

**Face personalization.**    To evaluate and showcase the capability of our DTI method in face personalization, we conducted experiments using randomly selected faces from the FFHQ dataset (Karras et al., 2019) as well as faces generated by DALL·E (Ramesh et al., 2021).

Since CrossInit specifically focuses on facial personalization, we compare TI, CrossInit and our DTI on this task. Given that CrossInit does not explicitly provide hyperparameters (including learning rate) tailored for SDXL, we performed a grid search across various learning rates. Our empirical results indicated that the learning rate used by TI yielded reasonable performance for CrossInit as well. Figure 14 illustrates a comparison between the three methods, demonstrating that all methods perform effectively for facial personalization. Nevertheless, as the complexity of text prompts increases (rows depicted in the left columns), the baseline methods struggle to accurately reflect all described components of the prompts. In contrast, our DTI method consistently captures the critical components precisely, demonstrating superior performance in achieving enhanced textual fidelity.

## E    ADDITIONAL EXPERIMENTS

We present additional experimental results in Figures 15, 16, 17, and 18. Specifically, Figure 15 compares TI using SLERP against LERP, justifying our choice of the latter. In Figure 16, we present an ablation study on magnitude settings. While our DTI uses the mean value of the entire vocabulary as the default, we further investigate initializing with the specific category describing the subject

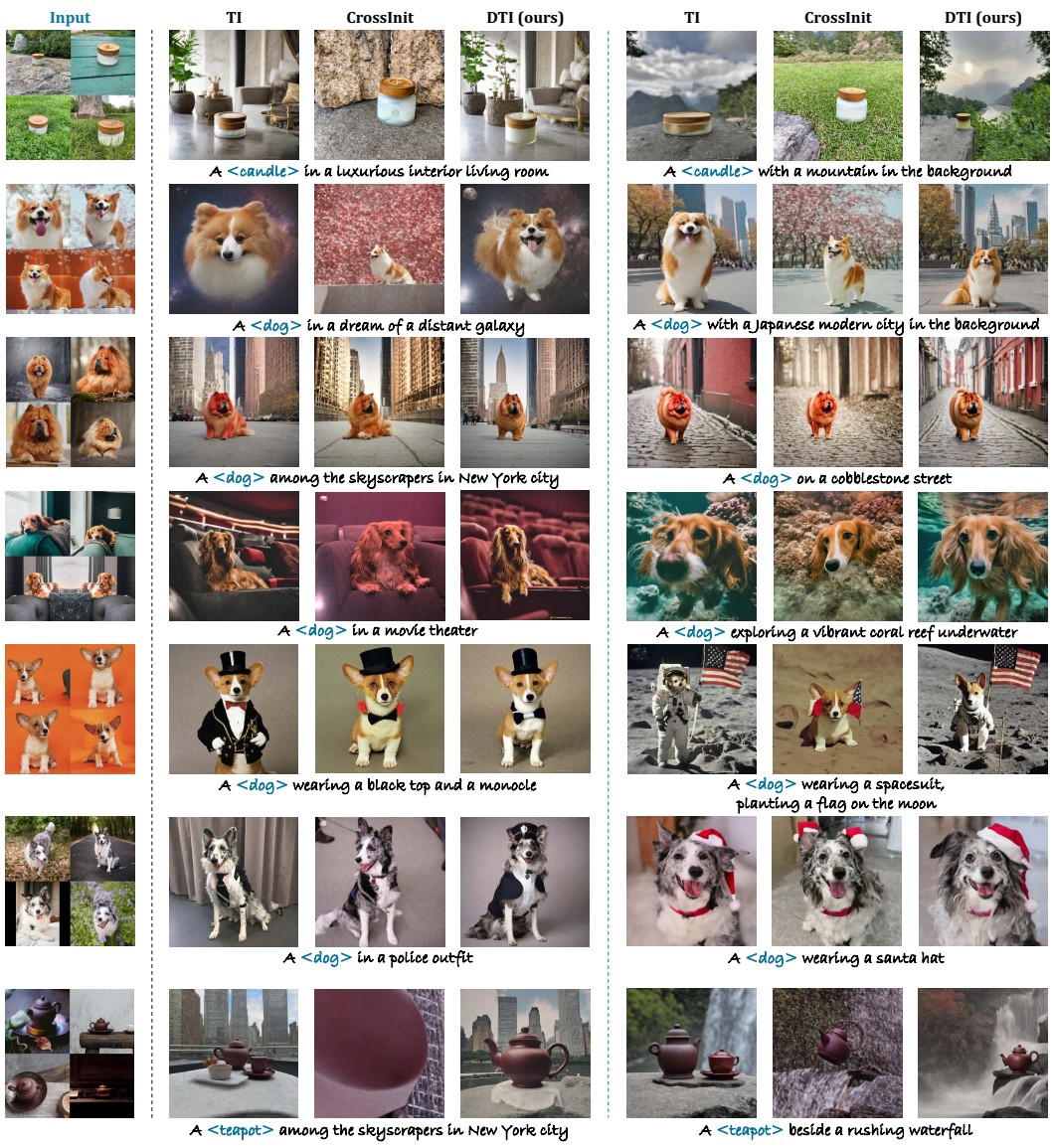

Figure 8: **Qualitative results with SDXL.** Here, we provide more qualitative comparisons with TI and CrossInit. Our DTI consistently generates results that precisely reflect the user text prompts, while maintaining subject similarity.

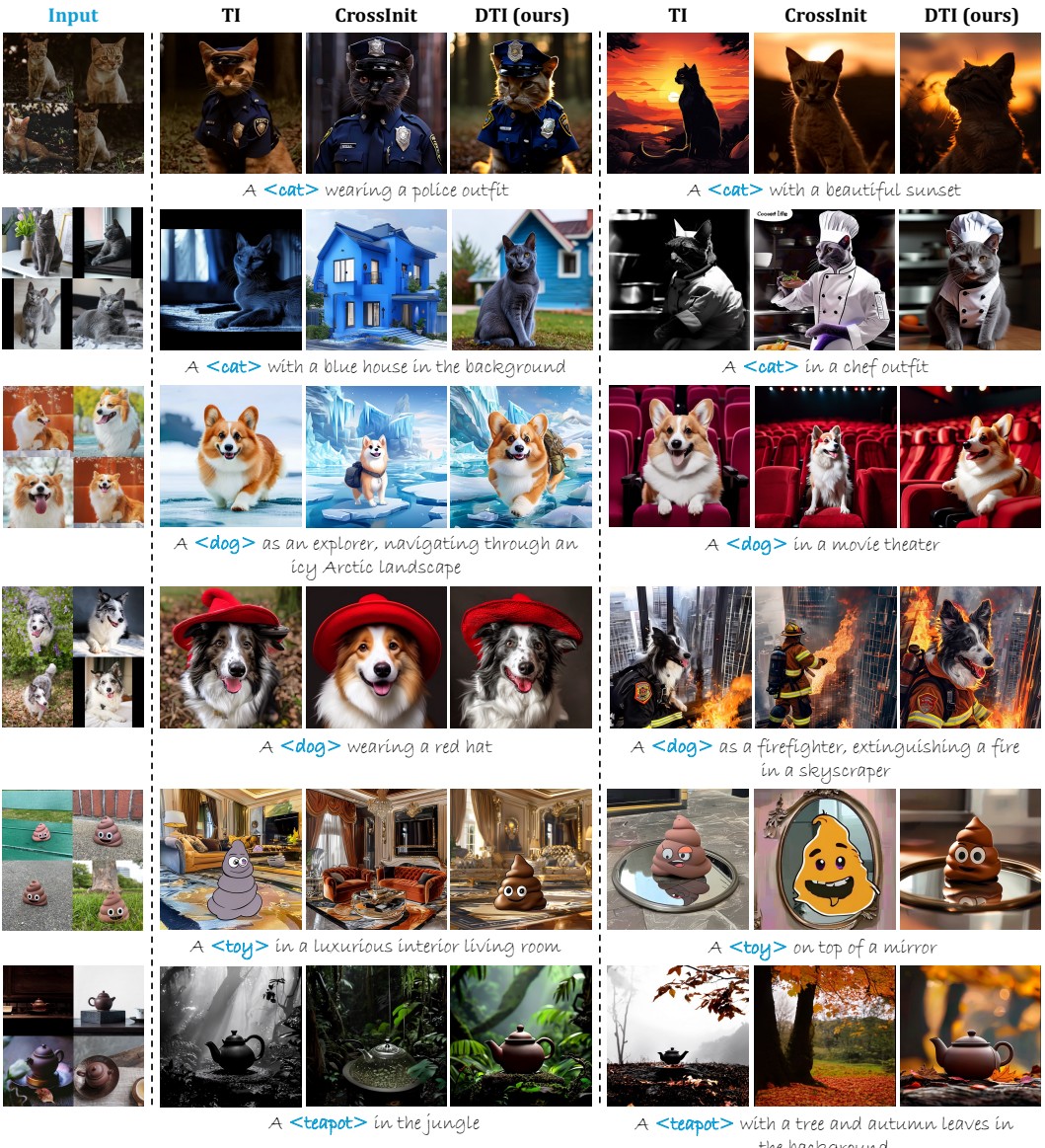

Figure 9: **Qualitative results with SANA1.5-1.6B.** Here, we provide more qualitative comparisons with TI and CrossInit on SANA. Our DTI consistently generates results that precisely reflect the user text prompts, maintaining subject similarity.

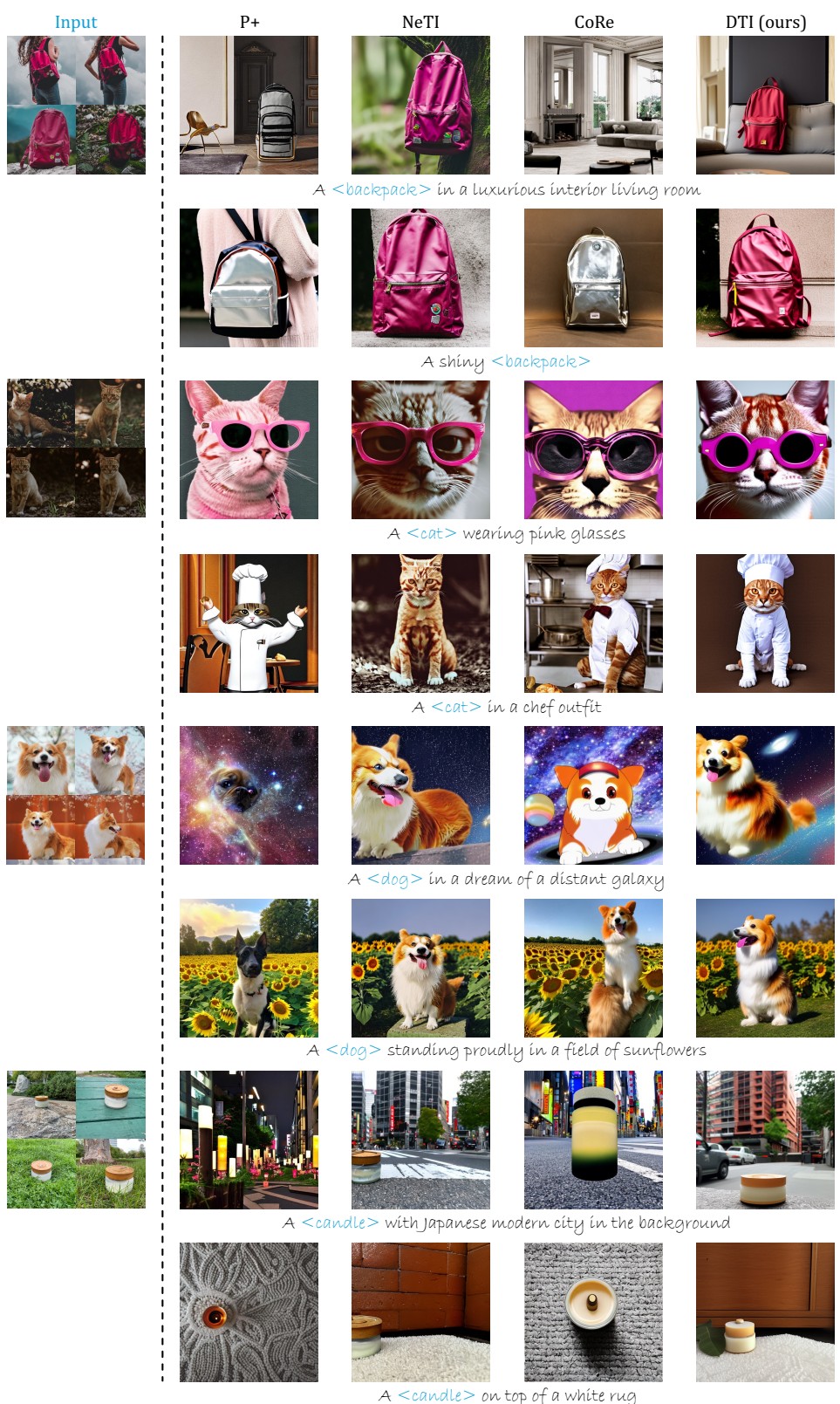

Figure 10: **Comparison with additional baselines.** We provide qualitative comparisons against additional TI-enhancing methods—P+, NeTI, and CoRe. Because these baselines are built on SD2.1-base, we apply DTI using the same pre-trained backbone to ensure fairness. The results demonstrate that DTI attains higher text fidelity while maintaining subject similarity.

**Input**

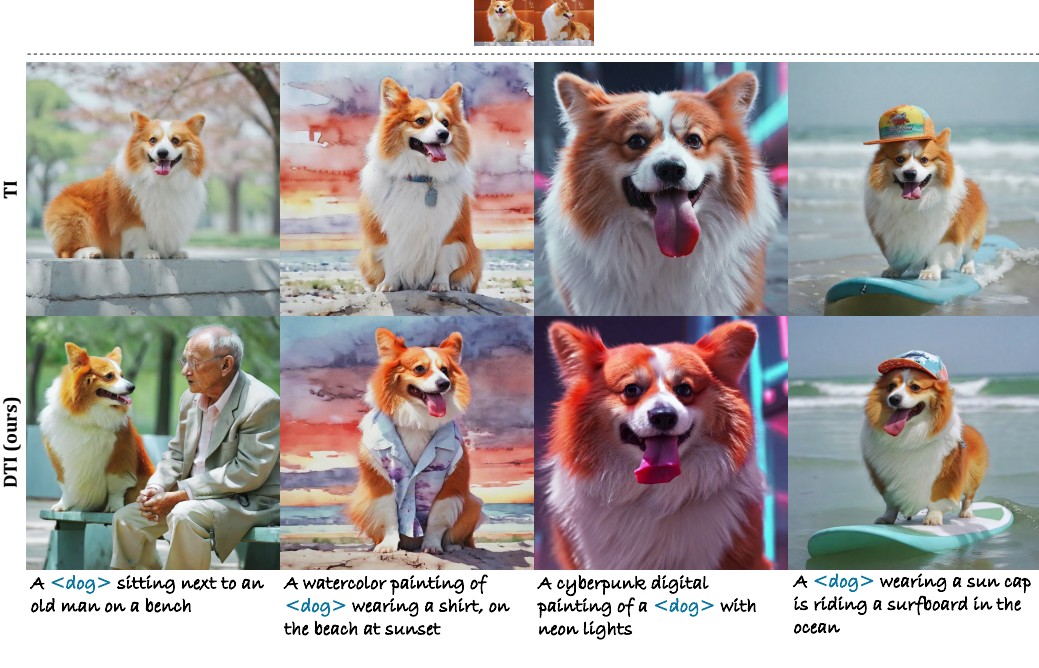

A \<dog\> sitting next to an old man on a bench

A watercolor painting of \<dog\> wearing a shirt, on the beach at sunset

A cyberpunk digital painting of a \<dog\> with neon lights

A \<dog\> wearing a sun cap is riding a surfboard in the ocean

**Input**

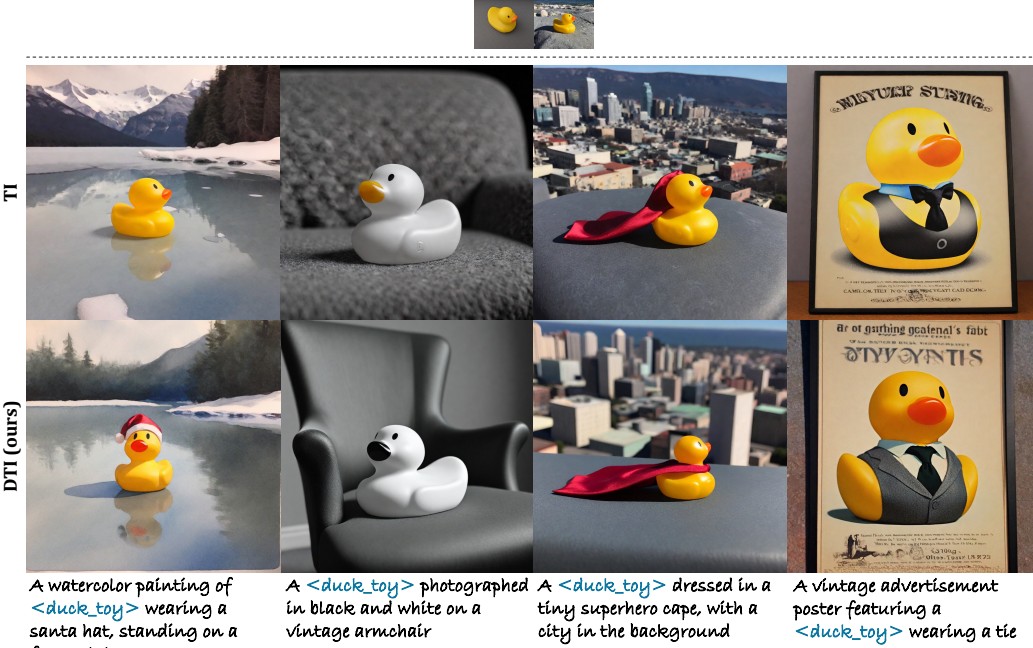

A watercolor painting of \<duck_toy\> wearing a santa hat, standing on a frozen lake

A \<duck_toy\> photographed in black and white on a vintage armchair

A \<duck_toy\> dressed in a tiny superhero cape, with a city in the background

A vintage advertisement poster featuring a \<duck_toy\> wearing a tie

Figure 11: **Qualitative results with TI/DTI with LoRA on SDXL.** We provide a qualitative comparison of TI and DTI when combined with LoRA-based fine-tuning (rank 32). DTI consistently improves the text prompt fidelity compared to TI.

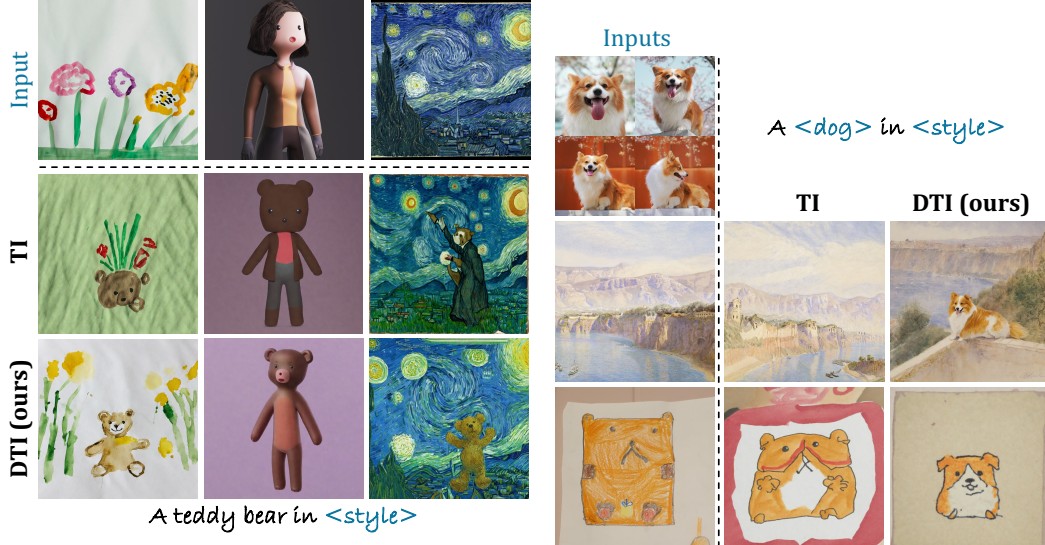

Figure 12: **Stylization.** Qualitative comparison of personalization with diverse style inputs.

Figure 13: **My subject in my style.** Qualitative comparison of subject-style mixing within the same prompt.

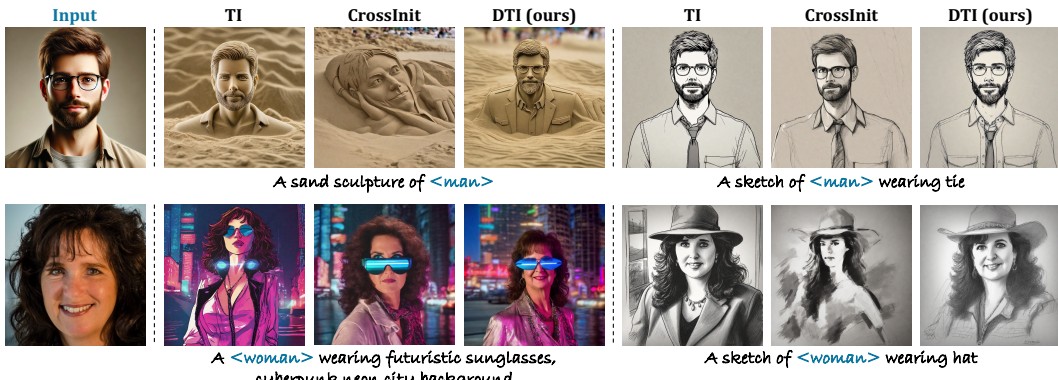

Figure 14: **Comparison of face personalization methods.** We compare our method and Textual Inversion (TI) against CrossInit, which specifically targets face personalization. To prevent bias from celebrity faces, we evaluate personalization using two alternative sources: images generated by DALL·E (Ramesh et al., 2021) (top row) and randomly selected images from the FFHQ (Karras et al., 2019) (bottom row).

(e.g., cat). We demonstrate that minor variations in magnitude do not significantly alter the outcome. Figure 17 evaluates our DTI in multi-concept scenarios, illustrating both successful outcomes and limitations. Finally, we analyze specific failure cases of our method in Figure 18.

## F  SOCIETAL IMPACTS

The rapid advancement of text-to-image diffusion models, especially in the domain of personalization techniques, raises important societal considerations. In particular, the ease of generating highly specific and detailed images can raise concerns related to copyright infringement, as personalized generative models may inadvertently or intentionally reproduce objects protected by intellectual property laws. Therefore, we note that it is important for users and distributors of the model to develop comprehensive awareness and implement guidelines addressing copyright boundaries, fair use, and ethical content generation. Moreover, we note that, since our method does not modify the

**SLERP (DTI)**

**Linear interpolation (TI)**

**Normalized → SLERP → multipled with dog's norm (TI)**

**Normalized → SLERP → multipled with teapot's norm (TI)**

<dog>                                                                <teapot>

Figure 15: **Interpolation options for TI.** We compare several interpolation options for TI, including linear interpolation, SLERP with normalization and adjusted norms. While these approaches exhibit minor differences in behavior, none produce smooth transitions between concepts. In contrast, our DTI with SLERP achieves noticeably smoother and more consistent interpolations.

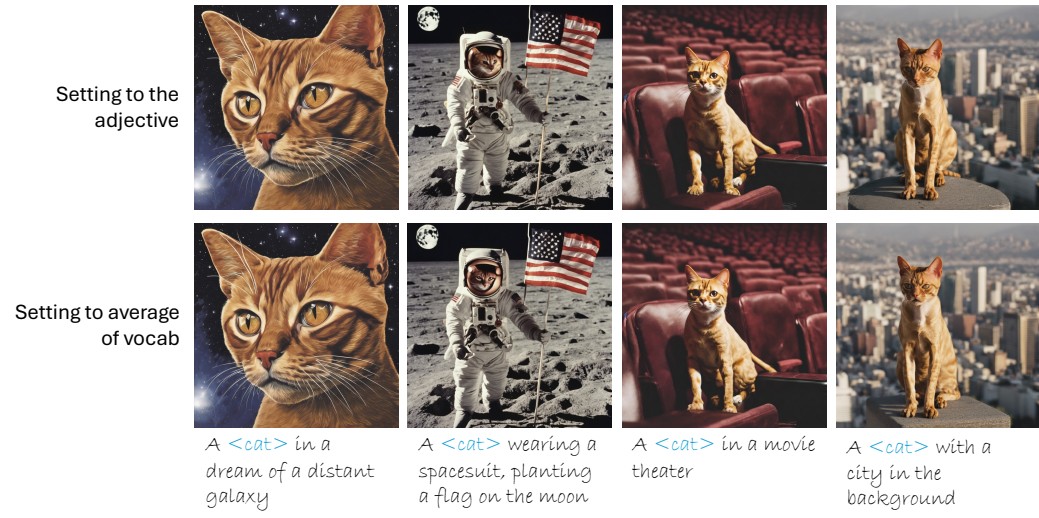

Setting to the
adjective

Setting to average
of vocab

A <cat> in a
dream of a distant
galaxy

A <cat> wearing a
spacesuit, planting
a flag on the moon

A <cat> in a movie
theater

A <cat> with a
city in the
background

Figure 16: **Ablation on magnitude settings.** For consistency and ease of use, all magnitudes in this paper are set to the average value computed over the model's vocabulary (see ablation in Table 3). To evaluate the effect of using concept-specific magnitudes (e.g., the magnitude of 'cat' for the concept <cat>), we provide ablation results under different magnitude settings. The results show that small deviations from the default magnitude do not lead to noticeable differences in output quality.

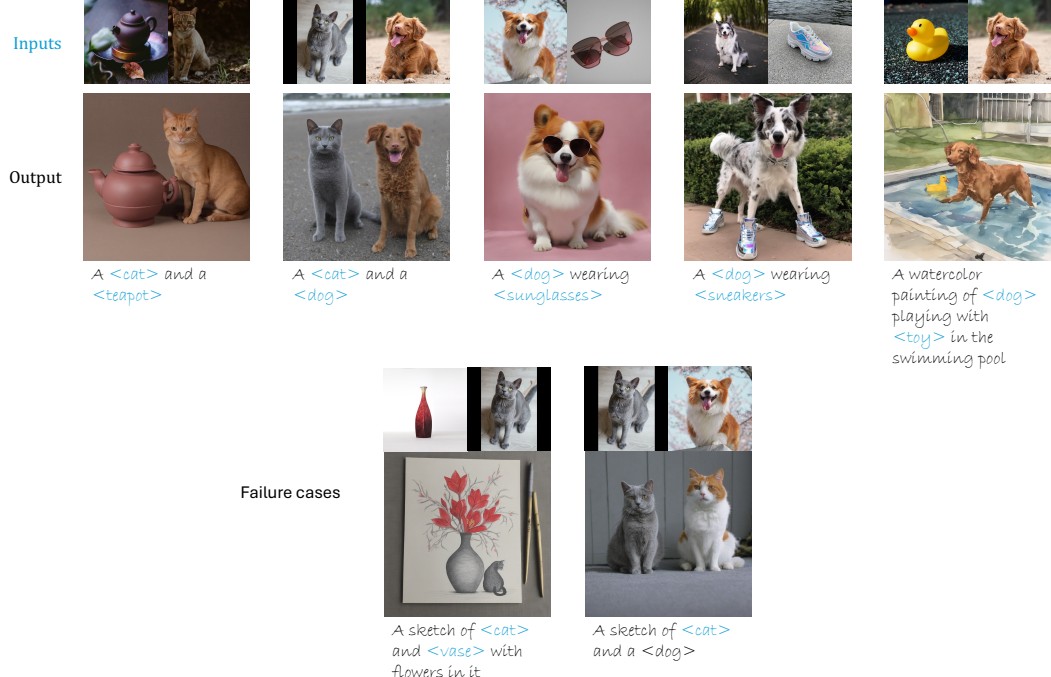

Inputs

Output

A \<cat\> and a
\<teapot\>

A \<cat\> and a
\<dog\>

A \<dog\> wearing
\<sunglasses\>

A \<dog\> wearing
\<sneakers\>

A watercolor
painting of \<dog\>
playing with
\<toy\> in the
swimming pool

Failure cases

A sketch of \<cat\>
and \<vase\> with
flowers in it

A sketch of \<cat\>
and a \<dog\>

Figure 17: **Multi-concept experiments.** We further evaluate DTI by combining multiple learned concepts within a single prompt. The results demonstrate that DTI can successfully integrate multiple concepts, while the second column shows failure cases exhibiting attribute binding issues.

underlying parameters of the generative model but solely adjusts the token embeddings that capture personalized concepts, the quality of generated images inherently depends on the capabilities of the underlying text-to-image model.

**When subject requires high visual detail**

Input

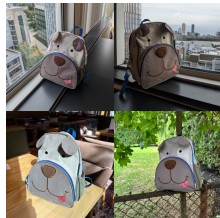
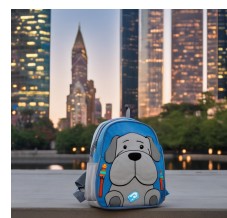

A \<backpack\> with a mountain in the background

A \<backpack\> on top of a wooden floor

A \<backpack\> with a futuristic cityscape in the background

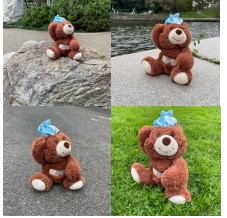
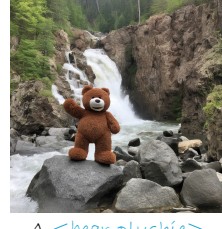
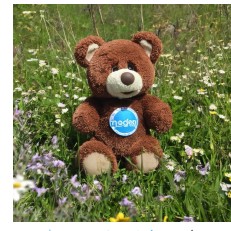
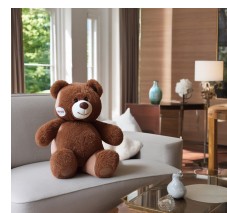

A \<bear_plushie\> besides a rushing waterfall

A \<bear_plushie\> in a field of wildflowers

A \<bear_plushie\> in a luxurious living room

**When prompt is difficult to depict**

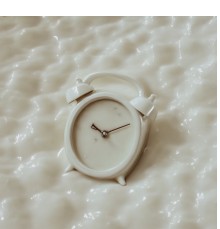
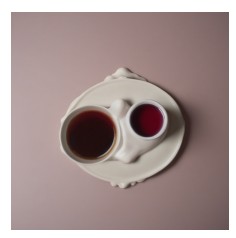
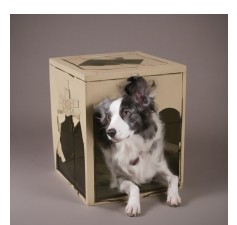
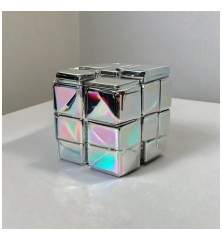

A \<subject\> floating in an ocean of milk

A cube shaped \<subject\>

**When prompt requires change of attributes**

☺  ☹  ☺  ☹

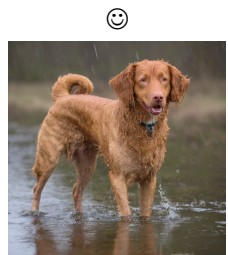
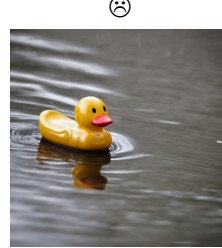
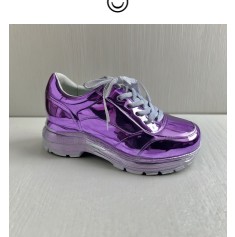
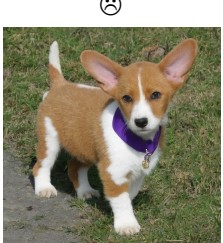

A wet \<subject\>

A purple \<subject\>

Figure 18: **Failure cases.** We present examples of three representative failure modes: (1) subjects that require high visual detail, (2) prompts that are vague or difficult to faithfully depict, and (3) prompts that involve attribute modification (e.g., color changes).

