# OpenReview forum: "Directional Textual Inversion for Personalized Text-to-Image Generation"
_ICLR.cc/2026/Conference — ICLR 2026 Poster_

### Official Review · Reviewer_r8HK · 2025-10-19

**Soundness:** 3
**Presentation:** 3
**Contribution:** 3
**Rating:** 8
**Confidence:** 4

**Summary:**

The paper addresses the text-to-image customization problem, leveraging a textual inversion method.

The authors showcase 2 aspects about textual inversion: the OOD magnitude of the inverted vector and the direction importance for the semantic meaning. Specifically, norm inflation or direction drift causes poor text alignment or object's poor fidelity in generated images.

As a solution, the authors propose to fix the magnitude of the inverted vector (aligned with the average in the vocabulary) and optimize only the direction.

**Strengths:**

1. Explanations for some of the known textual inversion problems (Effect I and Effect II).
2. Theoretical support of the claims (Proposition 1 and Corollary 1).
3. Despite some theoretical complexities, the method implementation is simple and flexible. No heavy finetuning required for diffusion models.

**Weaknesses:**

1. Qualitative comparisons are mostly or even only done with the baseline Textual Inversion method. Many improved or advanced methods are neglected in qualitative comparisons.
2. The proposed method's quantitative results are very close to the results of advanced textual inversion methods (NeTI, P+, etc), and the improvement is marginal.
3. There is no a limitations section or failed examples.

**Questions:**

1. Would you please provide more details about the norm inflation plot in Figure 1 (dataset, text encoders, etc.)?
2. In Proposition 1, there is a condition when $$||x^{(0)}|| > S_{L}$$
What if this condition is not met? What is the guarantee that the initial embedding's norm will be greater than the sum of all B's?
3. There is no description of S^{d-1} in Expression 1. Would you provide more details and descriptions about it?

---

> ### Author Response · Authors · 2025-11-21
>
> We thank the reviewer for thoughtful and constructive feedback. We appreciate your recognition of the strengths of our work, including the analysis of TI’s limitations, the theoretical support, and the simplicity and flexibility of the proposed method. We respond to each of your comments below:
>
> > **W1: More qualitative results with baselines other than TI**
>
> Thank you for the suggestion. In the revised version, we have added qualitative comparisons with P+ (Voynov et al., 2023), NeTI (Alaluf et al., 2023), and CoRe (Wu et al., 2025), which are provided in Figure 15 of the updated paper.
>
> In addition, we implemented DTI within the DCO framework (Lee et al., 2024). The corresponding quantitative and qualitative results are presented in Table 8 and Figure 5 of the updated paper. These results indicate that DTI can serve as a drop-in replacement for TI within model fine tuning pipelines, providing improved text fidelity while maintaining strong subject similarity.
>
> > **W2: Quantitative results with P+ and NeTI are marginal. What advantage?**
>
> While the quantitative gaps are narrow on SD1.5 and SD2.1-base (Table 7), we would like to note that our DTI achieves the best balance in both metrics. Furthermore, DTI offers critical advantages in **efficiency and architectural simplicity** compared to P+ and NeTI:
>
> - **1. Computational & Storage Efficiency:**
>   - P+ (Extended Textual Inversion): Replaces a single token with N distinct tokens (one per each layer). This increases storage and token management complexity by Nx.
>   - NeTI (Neural Textual Inversion): Replaces the token with a Neural Mapper network. Crucially, this network must be queried at every single timestep during inference, introducing significant computational overhead.
>   - DTI: Maintains the standard TI interface (1 vector, ~3KB), requiring no architectural changes or inference overhead.
>
> - **2. A Better Trade-off Curve:** DTI shows the best balance between text and image fidelity. On SD1.5, it achieves the highest subject similarity (0.418) while maintaining competitive text alignment, avoiding the severe trade-offs seen in P+ (which sacrifices subject fidelity) or NeTI (which sacrifices text alignment).
>
> - **3. Geometric Solution vs. Parameter Expansion:** Baselines improve performance by expanding the search space (adding layers or neural networks). DTI proves that the existing single-token space is sufficient if the optimization geometry is corrected (fixing norm inflation/stagnation). This geometric fix enables unique capabilities like SLERP-based interpolation (Figure 3), which is mathematically ill-defined in the unconstrained spaces of the baselines.

---

> ### Author Response · Authors · 2025-11-21
>
> > **W3: no limitation section or failed examples**
>
> Thank you for the suggestion. In the updated paper, we include Figure 15, where we show three types of failure cases:
>
> - **1. Subjects requiring high visual detail.**
>    When the target concept involves fine-grained structure or intricate textures, DTI may not capture all subtle details. This limitation is shared by TI-style methods, which compress the entire concept into a single embedding vector. As discussed in the paper, DTI is primarily designed to improve text prompt fidelity rather than subject reconstruction. However, because DTI is orthogonal to model fine tuning, it can be combined with lightweight methods such as LoRA (Figure 9) or DCO (Table 8 and Figure 5) to improve subject fidelity when needed.
>
> - **2. Difficult or ambiguous prompts.**
>    For prompts that are vague, conceptually ambiguous, or difficult to depict (e.g., requests such as "cube-shaped dog"), DTI may fail to produce the intended output. In these cases, the limitation often arises from the underlying text-to-image model’s capacity to interpret the prompt. DTI improves how a personalized token interacts with the prompt but cannot fully overcome inherent semantic ambiguity.
>
> - **3. Attribute-changing prompts.**
>    Prompts requiring substantial attribute changes of the personalized concept (e.g., drastic color or appearance modifications) yield mixed performance. DTI succeeds in certain cases but does not consistently handle all subjects.
>
>
> > **Q1: Figure 1 details**
>
> Figure 1 is computed on SDXL-base using its dual CLIP text encoders. In Figure 1(a), we plot the L2 norms of all vocabulary embeddings for each encoder (purple and orange histograms) and overlay the norm trajectory of the learned personalization token for the TI embedding of `<dog>` in the first row. As shown, TI pushes the token norm into an out-of-distribution range relative to the pretrained vocabulary.
>
> Figure 1(b) uses the second SDXL text encoder to visualize semantic drift. We collect the learned `<cat>` embeddings for both TI and DTI, gather several semantically related vocabulary tokens (e.g., 'cat', 'kitten', 'dog', 'gray', 'smile'), normalize all embeddings to unit norm, and project them to 2D using PCA. TI moves `<cat>` away from the 'cat/kitten' cluster, whereas DTI keeps it closer to its expected semantic neighborhood.
> Overall, Figure 1 illustrates both norm inflation and directional drift in SDXL’s dual encoders for a representative concept.
>
> > **Q2: What if $\||x^{(0)}\|| \leq S_L$?**
>
> Proposition 1 specifically targets the **large-norm regime** induced by TI. When $\||x^{(0)}\|| > S_L$, the cumulative rotation across $L$ blocks is tightly bounded, leading to directional freezing. Conversely, if $\||x^{(0)}\|| \leq S_L$, the global bound in Proposition 1 becomes less informative, but our per-block analysis in Lemma 2 still applies and shows that each pre-norm block retains a non-trivial capacity to rotate the vector.
>
> Moreover, this limitation can already appear in early layers. Applying the same argument as in Proposition 1 to the first $k$ blocks, the angular change after $k$ blocks satisfies
>
> $$
> \angle(x^{(0)}, x^{(k)}) \leq \frac{\pi}{2} \frac{S_k}{\||x^{(0)}\|| - S_k}, \quad S_k := \sum_{j < k} B_j.
> $$
>
> Thus, whenever $\||x^{(0)}\||$ is _significantly larger_ than $S_k$ (e.g., for $k=2$), the initial blocks already have very limited steering power, and subsequent blocks can only make small incremental updates to the direction. Since TI frequently induces strong norm inflation (Figure 1a), it often pushes the token into precisely this restrictive regime. DTI prevents this by keeping the embedding norm at an in-distribution scale, making the optimization focus on directionality and avoids directional freezing.
>
> > **Q3: What is $\mathcal{S}^{d-1}$?**
>
> We apologize for omitting the definition. Here, $\mathbb{S}^{d-1} =$ {$\mathbf{u} \in \mathbb{R}^d : \|| \mathbf{u} \||_2 = 1$} denotes the unit sphere. We have added this definition to our updated paper in Equation (1) of Section 3.1.
>
> We hope that our responses have addressed your concerns. We appreciate your constructive feedback and would be happy to provide further clarification if needed.

---

### Official Review · Reviewer_LKw4 · 2025-10-30

**Soundness:** 3
**Presentation:** 3
**Contribution:** 3
**Rating:** 6
**Confidence:** 5

**Summary:**

This paper focuses on enhancing personalized text-to-image generation by addressing critical limitations in Textual Inversion (TI), specifically the issue of embedding norm inflation that leads to poor prompt fidelity in complex scenarios. The authors first identify this bottleneck through empirical observations, such as the semantic drift and out-of-distribution magnitudes in learned tokens, and provide a theoretical analysis demonstrating how large norms attenuate positional information and cause residual stagnation in pre-norm Transformers. Building on this, they propose Directional Textual Inversion (DTI), a novel framework that fixes embedding magnitudes to in-distribution scales and optimizes solely the directional component on the unit hypersphere via Riemannian SGD, incorporating a von Mises-Fisher prior for semantic regularization.

**Strengths:**

The paper exhibits several notable strengths that enhance its scholarly contribution.

First, the analysis of Textual Inversion (TI) limitations is particularly insightful, as it introduces a new perspective by identifying and rigorously examining embedding norm inflation—a previously underexplored issue. This is supported by empirical evidence, such as the demonstration of out-of-distribution norms and semantic drift, and complemented by theoretical foundations that explain how large magnitudes disrupt Transformer dynamics, providing a comprehensive understanding of TI's failures.

Second, the technical pathway of Directional Textual Inversion (DTI) is well-justified and methodologically sound, building logically on the identified problems. By decoupling magnitude and direction, employing Riemannian SGD for hyperspherical optimization, and incorporating a von Mises-Fisher prior within a MAP framework, the approach effectively addresses the core issues, demonstrating a coherent and innovative design.

Third, the paper is exceptionally well-written, with clear organization and rigorous experimentation across diverse settings, including various personalization tasks, and comprehensive evaluations through metrics and human assessments.

**Weaknesses:**

The paper's introduction emphasizes TI's poor performance with complex prompts, yet the subsequent technical analysis in Section 2 appears decoupled from this specific issue. The theoretical framework primarily explains general performance degradation due to norm inflation but does not explicitly model or analyze the distinct challenges of compositional prompts, creating a slight disconnect between the stated motivation and the core technical solution.

While the paper claims TI is an "efficient alternative," this claim warrants critical examination. The requirement of approximately 7 minutes per subject for SDXL and 30 minutes for SANA, as mentioned in the reproducibility statement, along with TI's inherent limitation to optimizing a single concept per embedding, challenges its practicality for real-time applications or scenarios requiring multi-concept personalization. The efficiency argument is relative to full model fine-tuning but may not hold against more recent parameter-efficient methods.

The qualitative results, while generally supportive, exhibit inconsistencies that merit discussion. For instance, in the first column of Figure 2, the generated bears show noticeable variation in appearance, compromising subject consistency. Furthermore, the intended "wizard costume" attribute is not convincingly realized in the outputs, highlighting a potential limitation in capturing specific, fine-grained attributes through directional optimization alone.

A more comprehensive quantitative comparison is needed to fully situate DTI within the current landscape. The evaluation primarily benchmarks against standard TI and CrossInit. Including comparisons with a wider range of contemporary methods—such as other TI variants (e.g., P+, P*, NeTI), lightweight fine-tuning approaches like LoRA, or even full fine-tuning methods like DreamBooth on key metrics—would provide a clearer picture of DTI's relative advantages and trade-offs in terms of both performance and computational cost.

The theoretical analysis, though valuable, relies on the assumption of bounded sub-layers (Section B.1). The practical validity and tightness of these bounds in modern, large-scale text encoders are not empirically verified. Exploring how these bounds behave in practice could strengthen the theoretical claims or reveal conditions under which the effects of norm inflation are more or less pronounced.

The selection of the vMF prior's concentration parameter κ, while justified via a grid search, is presented as a fixed value (1e-4). The paper does not explore whether an adaptive κ, potentially based on training dynamics or the semantic specificity of the target concept, could lead to further improvements. This fixed hyperparameter approach might limit optimal performance across diverse personalization tasks.

**Questions:**

How does DTI's performance scale with the complexity and number of concepts in a single prompt?

Could the directional optimization framework be extended to multi-token representations?

---

> ### Author Response · Authors · 2025-11-21
>
> We thank the reviewer for thoughtful and constructive feedback. We sincerely appreciate your recognition of the strengths of our work, including the analysis of TI’s limitations, the study of norm inflation, the methodological soundness with its underlying theoretical motivation, the clarity of the writing, the comprehensive evaluation, and the inclusion of human assessment. We respond to each of your comments below:
>
> > **W1: Intro talks about TI’s poor performance with complex prompts, but the analysis talks about high norm. A slight disconnection here?**
>
> We appreciate this observation and agree it is important to connect the motivation to the theory more explicitly. In fact, the complex prompt failures in the Introduction and the analysis in Section  2 describe the same underlying phenomenon from complementary viewpoints.
>
> In the Introduction, we highlight that TI often fails on prompts that require integrating multiple attributes, backgrounds, or styles (e.g., the SDXL examples in Figure 1 and Figure 2 where TI omits 'santa hat', 'music stage', or 'Pop‑art style illustration of `<cat>`'). These are precisely cases where the text encoder must combine information from many tokens and positions.
> In Sec. 2.2, we analyze pre‑norm text encoders and show that large learned norms: (i) attenuate absolute positional embeddings as O(1/m), so the normalized representation becomes insensitive to where the token appears (Effect 1 / Lemma 1), and (ii) cause residual updates to stagnate, limiting the directional change of hidden states across layers (Effect 2 / Lemma 2, Proposition 1).
> Taken together, this gives a direct causal chain: **norm inflation → loss of positional/contextual signal + residual stagnation → failure to integrate multiple prompt constraints**, which manifests most strongly on complex prompts.
>
>
> > **W2: TI’s training time is too long compared to model-finetuning methods**
>
> We agree that efficiency should be interpreted with care. In our experiments, we follow the commonly used TI schedule of 500 training steps for both SDXL and SANA. Under this configuration, a wall-clock comparison on an A6000 GPU indicates:
> - TI / DTI (500 steps, SDXL): ~7 minutes per concept
> - Model fine-tuning (300 steps, SDXL, LoRA rank = 4): ~6 minutes
> These numbers suggest that, in our setup, the runtime difference is relatively modest. We also note that full DreamBooth-style fine-tuning typically requires generating 100–200 prior-preservation images, which may further increase the overall time depending on the implementation.
>
> Our aim in improving TI is not to argue that it is universally faster than parameter-efficient fine-tuning methods, but rather to highlight several practical aspects that remain appealing in many workflows:
> - Minimal per-concept storage (a single embedding vector rather than LoRA weights or full checkpoints).
> - No modification to the base model, enabling straightforward integration into existing pipelines.
> - No reliance on auxiliary prior-preservation datasets (as in DreamBooth).
>
> Finally, we provide additional results that DTI can also complement model fine-tuning approaches: combining DTI with DCO (NeurIPS’24) or LoRA tends to improve prompt fidelity while maintaining strong subject similarity (Table 8, Figure 5, and Figure 9).
>
> > **W3: Fig2 first column, bear lacks fidelity?**
>
> We agree that the bear plushie example in the first column of Figure 2 does not achieve perfect subject consistency. This limitation is expected, as TI-style methods (including DTI) optimize only a single embedding vector and therefore do not have the modeling capacity of full model fine-tuning approaches to reproduce highly detailed or structurally complex subjects.
>
> The purpose of Figure 2 is to compare TI-based methods under the same capacity constraint, rather than to suggest that single-token personalization can match the fidelity achievable with DreamBooth-style fine-tuning. Within this setting, DTI provides a more favorable trade-off: it tends to preserve more subject characteristics than TI and CrossInit while incorporating the requested prompt elements more reliably (e.g., background, composition, and style cues that TI and CrossInit often miss). This trend is reflected qualitatively in Figure 2 and quantitatively in Table 2, where DTI improves text fidelity while maintaining competitive subject similarity. We will add a clarifying sentence in the revised version to highlight this capacity limitation and the intended scope of the comparison.
>
> Finally, we note that DTI can also be combined with model fine-tuning methods. As shown in Table 8, Figure 5, and Figure 9, integrating DTI with LoRA improves prompt fidelity while further enhancing subject similarity.

---

> ### Author Response · Authors · 2025-11-21
>
> > **W4: more baselines?**
>
> In addition to the main paper comparisons with TI and CrossInit, we include several further baselines in Appendix D.3: P+ (Voynov et al., 2023), NeTI (Alaluf et al., 2023), and CoRe (Wu et al., 2025), on SD1.5 and SD2.1-base (Table 7), using the same metrics as in the main text. Across these settings, DTI achieves a competitive and often more balanced trade-off between subject similarity and text-prompt fidelity. We have also added qualitative comparisons in Figure 15.
>
> We further note that DTI can be integrated with model fine-tuning approaches. Specifically, we compare DCO + TI and DCO + DTI in Table 8 and Figure 5, where DCO + DTI attains higher prompt fidelity while maintaining strong subject similarity.
>
> > **W5: Section B.1’s boundness in practice?**
>
> We appreciate the reviewer for rigorously checking for our theoretical assumptions. We address the validity of the bound $B_l$ below.
>
> - **Theoretical justification:** The boundness of $B_l$ is not an approximation but a mathematical consequence of the architecture. Consider the domain of the sub-layer $F_l$, which is the image of the normalization layer:
> $\mathcal{S} = \{ \text{Norm}(z) : z \neq 0 \} $.
>
> - **Compactness:** In pre-norm Transformers, both LayerNorm and RMSNorm map any non-zero input to a bounded, closed, compact subset of $\mathbb{R}^d$ (e.g., vectors with fixed RMS or fixed variance and mean)
>
> - **Continuity:** Each $F_l$ consists of attention and MLP blocks, which are compositions of linear maps and point-wise non-linearities; hence, $F_l$ is continuous on $\mathbb{R}^d$. By the extreme value theorem, a continuous function on a compact set attains its supremum. Therefore, $B_l := \sup_{u \in \mathcal{S}} \|F_l(u)\|_2 < \infty,$ so the boundedness assumption always holds in practice.
>
> Intuitively, normalization layers enforce fixed variance or fixed RMS values at every block, which not only ensures mathematical boundedness but is also essential for numerical stability in large Transformer models.
>
> > **W6: Adaptive kappa?**
>
> We appreciate the reviewer for raising this point. Adaptive $\kappa$ is indeed a reasonable direction, and we implemented the suggested schedule. In this variant, $\kappa$ is initialized with a small value at the beginning of training and gradually increased so that the prior becomes more concentrated over time. The results are shown below:
>
> | κ setting     | Image | Text  |
> |---------------|-------|-------|
> | adaptive $\kappa$    | 0.476 | 0.490 |
> | fixed $\kappa$ (= 1e-4) | 0.450 | 0.522 |
>
> The two configurations yield broadly similar performance. Adaptive $\kappa$ provides slightly higher image similarity but lower text fidelity. Since DTI’s primary objective is to improve prompt alignment, we adopt the simpler fixed $\kappa = 1e-4$ configuration, which performs reliably across models and datasets while avoiding additional complexity. We thank the reviewer again for this constructive suggestion.

---

> ### Author Response · Authors · 2025-11-21
>
> > **Q1: Performance under complexity and number of concepts?**
>
> While Figure 7 focuses on simple prompts, Figure 2 demonstrates that DTI maintains both high text fidelity and strong subject similarity even under more challenging generation conditions. For multi-concept scenarios, Figure 11 provides results combining a subject token with a style token.
>
> To answer the reviewer’s question, we further conducted experiments involving multiple subject tokens, as shown in Figure 16. These results suggest that DTI can support such multi-concept compositions to a limited degree. However, we also observe instances where attribute binding across multiple learned tokens becomes inconsistent. We believe that fully resolving this issue will require additional structural mechanisms. Importantly, this challenge is orthogonal to the central contribution of DTI (analyzing and correcting the norm-inflation dynamics of TI at the single-concept level) and we view comprehensive multi-token personalization as an important direction for future work.
>
> > **Q2: Can it be extended to multi-token representations?**
>
> Yes, DTI naturally extends to multi-token representations. We empirically validated this in Appendix D.2 (Table 6), where we utilized Qwen-VL to recommend 1-2 descriptive initialization tokens instead of a single coarse class token.
>
> When using Qwen-VL, we initialized multiple directional tokens based on specific visual attributes identified by the VLM. For example, for `<cat>`:
> - Original: Single token initialized with 'cat'.
> - Qwen-VL Multi-Token: Two tokens initialized with 'gray' and 'cat'.
>
> As shown in Table 6, expanding the representation to multiple directional tokens (guided by VLM priors) improved subject similarity (from 0.450 to 0.520 on SDXL) by allowing the model to capture more nuanced visual features while maintaining the geometric constraints of DTI.
>
> We hope that our responses have addressed your concerns. We appreciate your constructive feedback and would be happy to provide further clarification if needed.

---

### Official Review · Reviewer_Z4f5 · 2025-10-31

**Soundness:** 2
**Presentation:** 3
**Contribution:** 3
**Rating:** 6
**Confidence:** 4

**Summary:**

This paper is related to the task of text-to-image personalisation generation. The authors use Textual Inversion as the base method and claim that one problem with this method is the high norm of embedding of learned concepts, which leads to semantic drift. To address this issue, the authors propose finding such embeddings on a unit hypersphere. Additionally, they propose using a von Mises–Fisher distribution as a prior to model the distribution of these embeddings.

**Strengths:**

1) The idea behind the presented method is clear, and the text is well written.
2) The paper provides a solid theoretical basis for the proposed modifications.
3) The authors demonstrate the additional capabilities of their method.
4) The paper contains many experiments and an extensive ablation study.

**Weaknesses:**

1) The work contains a lot of theoretical discussion, but most of it is based on asymptotical estimations (e.g. Lemmas 1 and 2, Proposition 1). The problems described do not seem obvious to me and I am not sure they could arise in practice. (questions 1, 2)
2) I am unsure about the correct evaluation of TI and CrossInit on SDXL (Table 2). It seems that the methods are slightly overfitted. (question 3)
3) Some parts of the work (e.g. Fig. 1, Tab. 7) lack details on how they were obtained. (questions 4, 5, 6)

**Questions:**

1) Could you demonstrate Effect 1 in practice and explain how your method deals with it? For example, you could train a classifier that takes the trained embedding after the Layer Norm of a particular concept as input and tries to predict the position of that embedding. You could train such a classifier for the base model, TI and DTI, and demonstrate the distribution of the resulting accuracies for different concepts.
2) Could you demonstrate Effect 2 in practice and explain how your method deals with it? For example, you could fix a concept and calculate the angle between the trained embedding and its initialization. You could show the distribution of different concepts for TI and your method.
3) Could the authors present the metrics for the earlier TI and CrossInit checkpoints from Table 2?
4) Could you please explain how you obtained Fig. 1b? What hyperparameters did you use to train the methods, and what was the checkpoint step?
5) Could you clarify the results in Table 7? Why are the results for your method all in bold when, as I can see, they are not always the best?
6) Could you clarify which settings were used to generate the images in Figure 2?

---

> ### Author Response · Authors · 2025-11-21
>
> We thank the reviewer for thoughtful and constructive feedback. We sincerely appreciate your recognition of the strengths of our work, including the clarity of the core idea, the quality of the writing, the solid theoretical foundations, the additional capabilities of our method (such as interpolation), and the extensive experiments with ablation studies. We respond to each of your comments below:
>
> > **W1, Q1: Show effect 1 in practice - For example, you could train a classifier that takes the trained embedding after the Layer Norm of a particular concept as input and tries to predict the position of that embedding. You could train such a classifier for the base model, TI and DTI, and demonstrate the distribution of the resulting accuracies for different concepts.**
>
> Following your suggestion, we directly tested whether increasing embedding magnitude makes positional information unrecoverable after the first pre-norm (scale-invariant) normalization (LN$_0$).
>
> We trained a 2-layer MLP classifier on the _frozen base text encoder_ to predict a token’s position in the sequence from the LN$_0$ output applied to (_token embedding + absolute positional embedding_).
> On the unmodified base encoder, the classifier achieves $\approx$ 100% accuracy, confirming that LN$_0$ retains enough information to recover position.
> We then froze the classifier and artificially scaled the norm of a single token before LN$_0$, sweeping through the magnitude $m = \{0.5, 1, 2, 4, 8, 16 \}$.
> Finally, we evaluated the classifier on TI-trained and DTI-trained embeddings.
>
> |         | Normal | m = 0.5 | m = 1 | m = 2 | m = 4 | m = 8 | m =16 | TI    | DTI   |
> |---------|--------|---------|-------|-------|-------|-------|-------|-------|-------|
> | Accuracy | 1.000  | 1.000   | 1.000 | 0.846 | 0.076 | 0.008 | 0.008 | 0.000 | 1.000 |
>
> As the token norm $m$ increases, positional prediction accuracy drops sharply.
> When evaluated on actual personalized tokens, TI collapses to near-zero positional accuracy, whereas DTI maintains accuracy comparable to the base model, since its magnitude stays in-distribution.
>
> This directly supports Effect 1: when $m$ is large, the normalized vector Norm($m\mathbf{v} + p$) becomes insensitive to the positional term $p$, causing positional cues to vanish (as predicted by Lemma 1). DTI avoids this failure mode by keeping magnitude in-distribution.
>
> > **W1, Q2: Show effect 2 in practice - you could fix a concept and calculate the angle between the trained embedding and its initialization. You could show the distribution of different concepts for TI and your method.**
>
> We would first like to clarify that Effect II (Lemma 2 / Proposition 1) formally describes directional changes of the hidden states inside a pre‑norm block during the forward pass, not the distance between the final learned embedding and its initialization. The latter is influenced by optimization dynamics (e.g., learning rate, number of steps, and prior strength) and is therefore only an indirect proxy for the residual “stagnation” analyzed in our theory. Nevertheless, we agree that your proposed measurement is a useful complementary diagnostic, and we report it below together with a more direct probe of Effect II inside the text encoder.
>
> Following your suggestion, we fixed each concept and measured the angle between its trained embedding and its initialization.
> Below we report three representative concepts (first three in alphabetical order) and the average over all 30 concepts:
> | Concept          | TI (°) | DTI (ours) (°) |
> |------------------|--------|----------------|
> | backpack         | 89.37  | 59.27          |
> | backpack_dog     | 86.82  | 58.90          |
> | bear plushie     | 90.12  | 50.52          |
> | **Average (all 30 concepts)** | **87.91** | **59.36** |
> The results show that TI’s learned embeddings deviate much more from their initialization than DTI’s.
>
> Since Effect II specifically concerns residual stagnation due to large norms within pre‑norm blocks, we further examined angular changes of the hidden states inside the text encoder. For each concept, we propagated the personalized token through the encoder and computed the angle between the hidden state before and after each block, then averaged these angles across all layers. Empirically, TI embeddings induce an average per‑block angular change of 21.33°, whereas our DTI embeddings achieve a larger average change of 33.52° (1.57x larger than TI). These results are consistent with the theory in Lemma 2: by keeping norms in‑distribution, DTI mitigates residual update stagnation and allows the encoder to make more substantial directional updates.

---

> ### Author Response · Authors · 2025-11-21
>
> > **W2, Q3: Table 2 seems overfitted. Previous checkpoints results?**
>
> To examine whether TI and CrossInit on SDXL might be overfitted at the reported checkpoint, we additionally evaluated earlier checkpoints (100, 200, 300, 400, and 500 steps), as shown below:
>
> | **Iteration** | **TI** Image | **TI** Text | **CrossInit** Image | **CrossInit** Text | **DTI (ours)** Image | **DTI (ours)** Text |
> |--------------:|-------------:|------------:|---------------------:|--------------------:|----------------------:|----------------------:|
> | **100**       | 0.377        | 0.417       | 0.285                | 0.592               | 0.413                 | 0.603                 |
> | **200**       | 0.457        | 0.359       | 0.415                | 0.571               | 0.460                 | 0.551                 |
> | **300**       | 0.495        | 0.320       | 0.476                | 0.515               | 0.472                 | 0.519                 |
> | **400**       | 0.547        | 0.289       | 0.525                | 0.466               | 0.468                 | 0.517                 |
> | **500**       | 0.561        | 0.292       | 0.545                | 0.464               | 0.450                 | 0.522                 |
>
> Across these checkpoints, we do not observe a strong indication of overfitting at step 500. TI and CrossInit show a gradual shift between image similarity and text fidelity, and DTI remains relatively stable throughout training. Overall, these results suggest that the checkpoint used in Table 2 is reasonably representative.
>
> > **W3, Q4: fig 1b setting**
>
> Figure 1b uses the same checkpoints as Table 2 for both TI and DTI. All hyperparameters follow Appendix D.1, and the same training configuration is applied consistently across all experiments.
> We note that Figure 1b is intended as a qualitative visualization of directional changes with respect to several representative vocabulary groups. Its purpose is to illustrate how TI and DTI move the concept embedding relative to nearby semantic directions, rather than to provide a detailed quantitative analysis of the full vocabulary space.
>
> For clarity, the visualization is generated through the following steps:
> (i) Load the learned embedding for the personalized concept (e.g., `<cat>` in Figure 1) along with embeddings from several semantically related vocabulary groups (animals, colors, emotions, etc.) in the SDXL text encoder, (ii) Normalize all embeddings to lie on the unit sphere, (iii) Apply PCA to project the embeddings into 2D, (iv) Plot the relative movement of the concept embedding under TI and DTI in this 2D space.
>
> For quantitative analysis, including exact angular measurements, please refer to our response to Q2.
>
> > **W3, Q5: table 7 result explanation, why DTI in bold?**
>
> You are correct that in Table 7 DTI is not the best on every individual metric. Our intention was to highlight the method that achieves the **best overall balance** between subject similarity and text fidelity across SD1.5 and SD2.1, not necessarily the column‑wise maximum. We apologize for the confusion and have adjusted the table formatting accordingly.
>
> > **W3, Q6: figure 2 setting**
>
> Figure 2 is generated using the same experimental settings as Table 2. We use a DDIM scheduler with 50 inference steps on SDXL, and all comparisons are conducted across multiple random seeds, with the seed kept fixed for each prompt to ensure fair comparison. We also noticed that the original caption did not explicitly mention SDXL, and have corrected it in the updated version of the paper. We apologize for the earlier omission. Additional qualitative results for both SDXL and SANA are provided in Appendix Figures 7 and 8.
>
> We hope that our responses have addressed your concerns. We appreciate your constructive feedback and would be happy to provide further clarification if needed.

---

### Official Review · Reviewer_CW52 · 2025-10-31

**Soundness:** 2
**Presentation:** 2
**Contribution:** 2
**Rating:** 4
**Confidence:** 4

**Summary:**

This paper proposes DTI, based on its key motivation that the norm of the special token inflates with training. They analyze the impact of this inflation by providing theoretical analysis on positional embedding and residual updates within the transformer. To mitigate this, they propose to learn only the direction with a vMF prior, while fixing the magnitude. The proposed method is claimed to yield improvements in text fidelity of TI and its variants.

**Strengths:**

1. It is useful that the norm inflation dilutes the impact of positional embedding and the residual updates within the transformer.
2. The claims are supported with theoretical justification.
3. The writing is clear and easy to follow.

**Weaknesses:**

1. The baselines are limited to heavily outdated methods. Further comparison is needed [1-3].
2. The norm inflation of the learned token had already been explored, not only in personalization [4], but also in VLM prompt tuning [5].
The quantitative comparison with [4] is provided, it is not clearly explain how the proposed approach differ conceptually.
3. Fig.3 - Is SLERP interpolation specifically available to the DTI? For fair comparison, SLERP should also be applied for TI for concept interpolation.
4. The choice of m* (average norm of the vocaubulary) is not justified. For example, m* can be the norm of the common adjectives e.g., yellow, fluffy, or the norm of the corresponding prior concept e.g., dog, cat.
5. Further comparison with recent optimizers e.g., Adam, AdamW should be provided to validate the effectiveness of the RSGD.


[1] Direct Consistency Optimization for Robust Customization of Text-to-Image Diffusion Models, ICLR'24
[2] Identity Decoupling for Multi-Subject Personalization of Text-to-Image Models, Neurips'24
[3] Controlling Text-to-Image Diffusion by Orthogonal Finetuning, Neurips'23
[4] Cross Initialization for Personalized Text-to-Image Generation, CVPR'24
[5] Nemesis: Normalizing the Soft-prompt Vectors of Vision-Language Models, ICLR'24

**Questions:**

1. The paper explains how the large norm disrupts the transformer's contextualization ability, but I'm curious why this happens. It would be more helpful to know why this phenomenon arises.
2. Should the norm always be excluded from learning? Could this be learned, but with regularization?

---

> ### Author Response · Authors · 2025-11-21
>
> We thank the reviewer for thoughtful and constructive feedback. We appreciate your recognition of our contributions in norm inflation analysis, theoretical justification, and writing clarity. Below, we provide additional experiments and clarifications in response to your comments.
>
> > **W1: Baselines are limited and outdated**
>
> We appreciate your suggestion. We would like to clarify the positioning of our baselines and the relationship between our method and fine-tuning approaches.
>
> **1. Relevance of current baselines:** We respectfully note that our chosen baselines are not outdated. We focus on embedding optimization, and the methods we compared, CrossInit (CVPR’24) and CoRe (AAAI’25), are among the most recent approaches in this specific regime. We select them to ensure a fair comparison within the same lightweight computational budget (~3KB storage, no weight updates).
>
> **2. Complementary with fine-tuning (DCO):** Methods you mentioned (DCO (NeurIPS’24), Orthogonal Fine-tuning (NeurIPS’23), and Identity-Decoupling (NeurIPS’24)) are model fine-tuning approaches, which operate in a heavier computational regime (GBs of storage, needs update in weights) compared to our embedding-only approach. However, we agree that exploring the intersection of these paradigms is valuable.
> Crucially, DTI is complementary to these fine-tuning methods. Since pipelines like DCO utilize Textual Inversion (TI) internally to achieve higher subject fidelity, DTI can serve as a drop-in replacement to enhance their performance. To demonstrate this, we integrated DTI into the DCO pipeline:
> | Method     | Image | Text  |
> |------------|-------|-------|
> | DCO        | 0.605 | 0.456 |
> | DCO + ours | 0.568 | 0.635 |
>
> As shown above, replacing standard TI with DTI in the DCO framework results in a significant gain in text fidelity (+0.179). While there is slight decrease in image similarity (-0.037), we argue this is a highly favorable trade-off for two reasons:
> **(i) Addressing the Bottleneck:** The primary failure mode of personalization is often the inability to follow complex prompts (semantic drift). A massive gain in text alignment fixes this critical bottleneck. We added qualitative comparison in Figure 5 of our updated paper.
> **(ii) Net Performance Gain:** When considering the average of both metrics (average score), DCO + DTI (0.602) substantially outperforms the baseline DCO (0.531).
> These results, which we have added to Appendix D.3 (Table 8 and Figure 5 of the revised paper), confirm that DTI provides a superior optimization capability even when used as a module within advanced fine-tuning pipelines.

---

> ### Author Response · Authors · 2025-11-21
>
> > **W2: Prior work has explored norm issues (CrossInit and Nemesis). How does DTI differ conceptually?**
>
> We clarify the conceptual differences between DTI and prior work such as CrossInit and Nemesis.
>
> - **Theory and design.** Prior work such as CrossInit and Nemesis is largely motivated by _empirical observations_ that TI embeddings may drift in scale or direction, and they introduce heuristic losses or initialization strategies to counteract this. In contrast, our approach begins with an explicit **theoretical analysis** of pre-norm text encoders: we show that large learned-token norms (i) attenuate absolute positional embeddings and (ii) cause residual updates to stagnate, leading to directional “freezing” (Proposition 1 / Corollary 1). Based on these results, DTI fixes the embedding magnitude to an in-distribution value and optimizes only the direction on the unit sphere using a vMF prior and Riemannian updates (Algorithm 1), directly addressing the identified failure mechanisms.
>
> - **CrossInit vs. DTI.** CrossInit is primarily driven by empirical observations: it initializes the personalized token using the encoder output and adds a consistency loss that keeps the learned embedding close to this output. In our SDXL setup, these encoder outputs have large norms, and the consistency constraint keeps the token embedding direction closely aligned with the output embedding. Thus, CrossInit implicitly maintains both large magnitude and similar direction throughout training. In contrast, DTI is built from an explicit analysis of pre-norm encoder behavior: we avoid this large-norm regime by fixing the magnitude to an in-distribution value and optimizing only the direction on the hypersphere (vMF prior + Riemannian updates), directly addressing positional attenuation and residual stagnation.
>
> - **Nemesis vs. DTI.** Nemesis likewise starts from empirical observations about soft-prompt behavior for VLM classification. It introduces losses that regularize norms but still keeps both direction and magnitude learnable with Euclidean updates followed by a selection step. DTI, by contrast, targets text-to-image personalization for pre-norm encoders and enforces a fixed magnitude with direction-only optimization on the hypersphere using a vMF prior, derived directly from our theoretical analysis.
>
> While CrossInit and Nemesis are empirically motivated methods that stabilize or regularize embedding norms, **DTI provides a theoretically grounded analysis** of pre-norm encoders and then designs a solution that removes the norm from the optimization entirely.

---

> ### Author Response · Authors · 2025-11-21
>
> > **W3: Why not SLERP with TI?**
>
> SLERP performs interpolation on a sphere and therefore assumes that all parameters lie on a constant-radius manifold. DTI enforces exactly this geometry: concept embeddings are parameterized on a fixed-norm hypersphere as defined in Equation 1. Consequently, SLERP between two DTI concepts follows a natural geodesic in the model’s parameter space and remains within the constraint set by construction.
>
> In contrast, TI embeddings are unconstrained and exhibit substantial variation in their norms, often lying far outside any consistent manifold. For example, in the first row of Figure 3, the learned embedding for `<dog>` has a norm of 22.77, whereas the embedding for `<teapot>` has a norm of 16.00. Because TI offers no canonical spherical geometry, applying SLERP requires additional design choices, such as whether to normalize embeddings or how to handle their differing radii.
>
> Therefore, we follow standard practice for TI-style methods and use linear interpolation in Euclidean embedding space. We additionally evaluated a TI-SLERP variant in which the two TI embeddings are normalized prior to interpolation, where corresponding results are provided in Figure 13 using the same instances and seed as in Figure 3. These results show that, even with this normalization-based SLERP variant, TI interpolations remain substantially less coherent than DTI, and alternative SLERP design choices do not remedy this gap.
>
> > **W4: Why m\* to be avg of vocab? How about m\* with common adjectives or prior concepts**
>
> Our choice to set $m^∗$ to the average norm of the vocabulary is supported by both theoretical considerations and empirical evidence.
> (1) As discussed in Section 2, semantic information in token embeddings is primarily encoded in direction. Embeddings with excessively large norms weaken semantic conditioning by diminishing positional signals and causing residual stagnation. Thus, maintaining magnitudes within the in-distribution range of the pretrained vocabulary is crucial. Using the mean vocabulary norm naturally satisfies this constraint.
> (2) This choice also offers a practical benefit: it avoids manual or heuristic selection of adjectives or auxiliary tokens and provides a stable initialization that generalizes across concepts. Table 3 demonstrates its effectiveness.
> To address the suggested alternative, we also evaluated initializing $m^∗$ using common adjectives or semantically related prior concepts (e.g., ‘orange’ for `<cat>`, ‘dog’ for `<dog>`). Applied across all 30 concepts, the averaged results are:
>
> | Initialization strategy            | Image | Text  |
> |------------------------------------|-------|-------|
> | Using common adjectives            | 0.463 | 0.514 |
> | Average of whole vocabulary (ours) | 0.450 | 0.522 |
>
> The adjective-based strategy yields marginally higher image similarity but slightly lower text fidelity, offering no clear advantage while requiring additional manual selection. Qualitative results in Figure 14 further show minimal visual differences.
> In summary, while adjective-based initialization is a reasonable option, using the vocabulary-average norm provides comparable performance, without necessitating manual setting of tokens.

---

> ### Author Response · Authors · 2025-11-21
>
> > **W5: Further comparison with recent optimizers**
>
> We note that we have compared our method with AdamW optimizer in Table 3. Below are the results of the comparison with the AdamW optimizer. The results show that RSGD yields higher image and text fidelity, supporting our claim that respecting the hyperspherical geometry is beneficial.
>
> | Optimizer      | Image | Text  |
> |----------------|-------|-------|
> | AdamW          | 0.335 | 0.463 |
> | RSGD (ours)    | 0.450 | 0.522 |
>
> > **Q1: Why does norm inflation arise during TI training?**
>
> Thank you for your question, we agree that it is an important point to clarify.
>
> In standard TI, the new token is optimized in ordinary Euclidean space with AdamW, and the loss does not include any term that keeps its norm small. In a pre-norm encoder, however, the network almost ignores this overall scale: LayerNorm/RMSNorm largely remove the magnitude before each attention or MLP block, so changing the token from $m\mathbf{v}$ to $\alpha m \mathbf{v}$ only weakly changes the normalized input (Sec. 2.2, Lemma 1).
>
> Because of this, increasing the norm is an easy direction for the optimizer to move in: gradients naturally have some component that points outward, and nothing in the objective pulls back toward the vocabulary range. Over training, these outward components add up, the token norm grows far beyond typical vocabulary norms, and we observe the norm inflation shown in Figure 1a. In our analysis (Lemma 2 / Proposition 1), we further show how such large norms can make residual updates very small, which helps explain why these inflated embeddings hurt text conditioning.
>
> > **Q2: Could the norm be learned with regularization? (like Nemesis?)**
>
> Thank you for the insightful question. Yes, it is possible to keep the norm learnable and introduce a regularization term that encourages it to remain within the typical vocabulary range. This idea is indeed related to Nemesis, which employs norm-based penalties while jointly optimizing both magnitude and direction in Euclidean space.
>
> Our objective in this work, however, is slightly different. From our analysis of pre-norm encoders, we find that as long as the embedding norm stays within the in-distribution vocabulary scale, the semantic information is primarily encoded in the direction, whereas large norms mainly exacerbate the two issues we identify: positional attenuation and residual stagnation. Motivated by this observation, we adopt a simpler formulation: we fix $m^∗$ to an in-distribution value and optimize only the direction on the sphere.
>
> This design provides several practical advantages:
> (i) it avoids introducing additional hyperparameters for norm penalties, (ii) it yields stable behavior across models and datasets (supported by the $m^*$ and $\kappa$ ablations in Table 3 and Appendix D.4), and (iii) it results in a clean geometric structure that naturally supports SLERP-based interpolation.
> While incorporating learnable norms with regularization is an interesting direction for future work, it is orthogonal to our primary goal in DTI, which is to simplify the optimization by removing unnecessary degrees of freedom.
>
> We hope that our responses have addressed your concerns. We appreciate your constructive feedback and would be happy to provide further clarification if needed.

---

### Author Response · Authors · 2025-12-03
**General response**

We sincerely thank all reviewers for their careful evaluation of our submission. We appreciate the constructive and detailed feedback, which has helped us further clarify and strengthen our work. Across the reviews, we truly appreciate the consistent recognition of:
- **A theoretically grounded analysis** of embedding norm inflation in Textual Inversion, supported by both empirical and formal results.
- **The conceptual and practical simplicity of our DTI** as a drop-in replacement for TI, while requiring no architectural changes and significantly enhancing text fidelity.
- **The clarity and comprehensiveness of the empirical evaluation**, including multi-model experiments, ablations, quantitative and qualitative experiments, and creative application on concept interpolation.

With further discussion unavailable due to this year’s OpenReview issue, we summarize below the significance of our work and the key revisions made.

In response to the reviews, we have:
- Demonstrated that DTI acts as a drop‑in replacement for TI inside recent fine‑tuning pipelines by integrating it into DCO. This substantially improves text fidelity while maintaining strong subject similarity (Table 8, Fig. 5).
- Added more comparisons with additional baselines. While quantitative comparisons with P+, NeTI, and CoRe were already included in the original submission (Table 7), we have now added qualitative comparisons on SD1.5 and SD2.1-base (Fig. 15). These additional visual results help illustrate how DTI achieves a stronger trade-off between subject similarity and text alignment among embedding-only methods.
- Added empirical supports of our theoretical analysis:
  - For Effect I (positional attenuation), we trained a classifier to recover token positions from the first pre‑norm output and showed that accuracy collapses as norms grow, matching our theory. TI‑trained embeddings fall into this regime, while DTI stays in‑distribution and preserves positional information.
  - For Effect II (residual stagnation), we measured angular changes of hidden states across layers and showed that large‑norm TI embeddings lead to smaller per‑block turns than DTI, confirming our analysis.
Expanded limitations and failure cases (Fig. 17) and discussed regimes where single‑token personalization is insufficient (fine‑grained subjects, highly ambiguous prompts, and strong attribute changes).
- Added multi-concept personalization experiments (Fig. 16)
- Provided additional ablations and clarifications on (i) the choice of fixed magnitude $m^*$ (table to reviewer CW52, Fig. 14), (ii) fixed vs adaptive $\kappa$ for the vMF prior, and (iii) Riemannian vs Euclidean optimizers on the sphere.
- Clarified how we generated all qualitative figures (including SDXL vs SANA settings) and how TI‑SLERP variants behave compared to DTI‑SLERP (Fig. 13).

We believe these additions address the main concerns about (1) the practical relevance of the theory, (2) the breadth of baselines, and (3) the design choices in DTI (norm fixing, prior strength, and optimization on the hypersphere). We hope the revised version and additional experiments make our contributions clearer and demonstrate that our DTI provides an effective and scalable way to improve prompt fidelity while keeping the simplicity and efficiency of embedding‑based personalization.

Once again, we sincerely thank all reviewers and the area chair for their time, thoughtful evaluations, and constructive feedback.

---

### Meta-Review · Area_Chair_t6N4 · 2026-01-05

**Summary:**

The authors propose Directional Textual Inversion (DTI) to address a key weakness of standard Textual Inversion, embedding norm inflation, which harms prompt fidelity in pre-norm Transformers. By fixing the embedding magnitude and optimizing only the direction on the unit hypersphere, DTI preserves semantic structure and avoids the drift seen in TI. Most reviewers find that the paper offers a clear and well-supported analysis of TI’s limitations, especially embedding norm inflation, and most claims are well grounded in sound theoretical justification.

**Reviewer Concerns:**

Several concerns were raised, such as the use of outdated baselines, prior work already discussing norm inflation, and marginal quantitative improvements, etc. After reading the authors’ rebuttal, the AC finds that most concerns have been adequately addressed.

**Reviewer Scores:**

This paper receives the following ratings: marginally below, marginally above, marginally above, and accept.  If they had been able to participate fully in the discussion, I would expect that marginally below rating will be increased, e.g., to marginally above. The AC recommends accepting this paper.

---

### Decision · Program_Chairs · 2026-01-26

Accept (Poster)